# SetPO: Set-Level Policy Optimization for Diversity-Preserving LLM Reasoning

Chenyi Li [*][1]  Yuan Zhang [* ♠][2]  Bo Wang [3]  Guoqing Ma [2]  Wei Tang [2]  Haoyang Huang [2]  Nan Duan [2]

## Abstract

Reinforcement learning with verifiable rewards has shown notable effectiveness in enhancing large language models (LLMs) reasoning performance, especially in mathematics tasks. However, such improvements often come with reduced outcome diversity, where the model concentrates probability mass on a narrow set of solutions. Motivated by diminishing-returns principles, we introduce a set level diversity objective defined over sampled trajectories using kernelized similarity. Our approach derives a leave-one-out marginal contribution for each sampled trajectory and integrates this objective as a plug-in advantage shaping term for policy optimization. We further investigate the contribution of a single trajectory to language model diversity within a distribution perturbation framework. This analysis theoretically confirms a monotonicity property, proving that rarer trajectories yield consistently higher marginal contributions to the global diversity. Extensive experiments across a range of model scales demonstrate the effectiveness of our proposed algorithm, consistently outperforming strong baselines in both Pass@1 and Pass@K across various benchmarks.

## 1. Introduction

The field of reinforcement learning (RL) has recently undergone a rapid expansion, with particularly strong advances in mathematical reasoning and code generation. This progress is largely driven by reinforcement learning with verifiable rewards (RLVR), whose scalability enables effective training at scale, thereby pushing the reasoning capabilities of large language models (Ouyang et al., 2022; Guo et al., 2025).

Within this paradigm, group relative policy optimization (GRPO) (Shao et al., 2024) emerged as a competitive alternative to proximal policy optimization (PPO) (Schulman et al., 2017), achieving strong empirical performance while eliminating the need for a critic model. This design choice has inspired a line of subsequent studies that build upon GRPO to improve training stability and empirical performance across diverse benchmarks (Yu et al., 2025; Zheng et al., 2025; Zhao et al., 2025).

While recent research has primarily focused on developing more effective group-based policy optimization algorithms, another line of work has observed that the apparent performance gains achieved through reinforcement learning often come at the cost of reduced model diversity. More concretely, recent evidence suggests that many trajectories produced by RLVR finetuned models can also be recovered from the base model by increasing the sampling budget, implying that the base model may act as an effective upper bound under extensive rollouts (Yue et al., 2025). Similarly, concurrent work in the LLM literature has also found that policy performance improvements are often accompanied by a decline in policy entropy, suggesting that enhanced performance may result from focusing the model's output distribution on a narrower set of solutions (Cui et al., 2025).

These observations highlight effective exploration as a key missing ingredient in RLVR based post-training, which is crucial for maintaining diverse outcomes while improving reasoning performance. However, unlike traditional reinforcement learning with a limited state-action space, the policy of LLMs, with their extended vocabulary size, operates in a high-dimensional space. As a result, the search space grows exponentially with the sequence length, imposing a significant burden on the exploration process (Gupta et al., 2024). Existing solutions often rely on entropy based regularization as a proxy for exploration in RL (Yao et al., 2025; Cheng et al., 2025). However, when entropy is defined at the token level and aggregated over sequences, it can introduce a strong length bias. Longer responses naturally accumulate higher entropy due to increased token-level uncertainty, thereby favoring verbose outputs regardless of whether they provide genuinely diverse solutions. Moreover, token level entropy fails to capture semantic diversity at the trajectory level, as it measures local uncertainty in next-token prediction rather than global variation in the meaning of complete

---

[*]Equal contribution ♠Project Leader [1]Peking University [2]JD Explore Academy, China [3]Beijing Institute of Technology. Correspondence to: Chenyi Li <lichenyi@stu.pku.edu.cn>, Yuan Zhang <zhangyuan.430@jd.com>, Nan Duan <duannan@jd.com>.

*Proceedings of the 43rd International Conference on Machine Learning*, Seoul, South Korea. PMLR 306, 2026. Copyright 2026 by the author(s).

## Evolution of Trajectories Embeddings

**GRPO: trajectory embeddings**     **SetPO: trajectory embeddings**

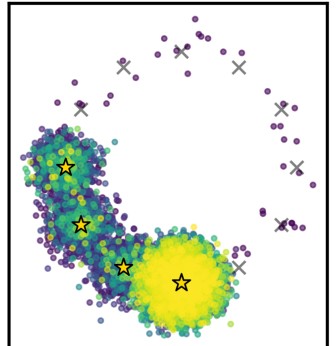 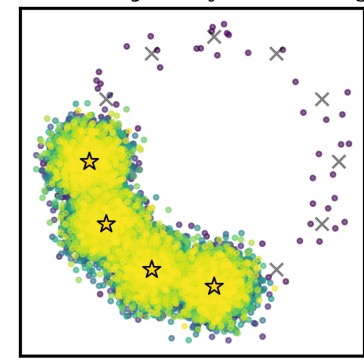 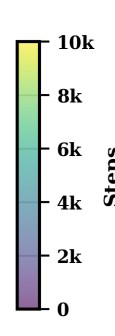

*Figure 1.* Evolution of trajectory embeddings during training for GRPO vs. SetPO (stars denote correct modes; color indicates training steps). Left (GRPO): Although trajectories initially populate multiple modes, training progressively concentrates on a single dominant cluster, exhibiting clear mode collapse. Right (SetPO): Embeddings remain distributed across multiple clusters throughout training, continuing to cover the correct modes at late steps. This indicates that SetPO mitigates mode collapse and preserves rollout diversity.

outputs. In addition, some recent studies explicitly optimize for Pass@K (Chen et al., 2025b; Tang et al., 2025; Walder & Karkhanis, 2025). Yet, since Pass@K primarily rewards success under repeated sampling, such improvements may not translate into higher single-sample correctness, leading to limited gains or even degraded performance in Pass@1.

In this paper, we propose SetPO, a diversity-preserving policy optimization algorithm that encourages the learned policy to represent a multi modal outcome distribution. SetPO assigns each trajectory a set-level marginal diversity credit, yielding an explicit diversity-aware signal for policy optimization. Concretely, for each prompt, we sample a group of rollouts and compute pairwise semantic similarities using a pretrained embedding model. Based on these similarities, we quantify how redundant each trajectory is within the group and define its diversity contribution via a leave-one-out marginal estimator. Trajectories that cover distinct semantic modes receive higher credits, whereas redundant ones receive lower credits. We then incorporate this set-level diversity signal into standard reward maximization by augmenting the per-trajectory training advantage with a weighted diversity term. To validate the effectiveness of the proposed algorithm, we construct a toy multi-modal bandit environment with twelve discrete semantic modes, where only the first four modes yield positive reward. Each mode is represented by a 50-dimensional embedding, and semantic similarity is measured via cosine similarity. As shown in Figure 1, our method preserves more distinct modes throughout training than vanilla GRPO. Consequently, it mitigates mode collapse and produces a more diverse set of outcomes.

Our main contributions are listed as follows.

- To the best of our knowledge, SetPO is the first to introduce an explicit set-level diversity objective. Dis-

tinct from traditional pairwise methods, our approach adopts a distributional perspective: it evaluates the holistic diversity of the entire sampled set by aggregating semantic similarities, rather than focusing on isolated local differences.

- We incorporate a leave-one-out marginal contribution mechanism into policy optimization, where each trajectory is rewarded based on its impact on the set's diversity. Our theoretical analysis demonstrates the anti-redundant nature of this estimator, showing that it assigns higher marginal value to rarer samples.

- We evaluate the proposed SetPO algorithm on comprehensive reasoning benchmarks, spanning model scales from 1.5B to 32B. Empirical results demonstrate that our method consistently achieves superior performance compared to strong baselines.

## 2. Related Works

**Reinforcement Learning for LLM Reasoning.** A series of studies (Zhao et al., 2025; Liu et al., 2025; Shrivastava et al., 2025) have been proposed to improve GRPO-style optimization for RLVR. DAPO (Yu et al., 2025) introduces several modifications to enhance training stability and sample efficiency. In particular, it adopts asymmetric clipping by using distinct clip ratios $\epsilon_{low}$ and $\epsilon_{high}$, allowing low probability tokens to receive effective updates. Moreover, DAPO filters out prompts for which all sampled trajectories are correct or incorrect, thereby focusing optimization on samples that provide meaningful learning signals. In addition, GSPO (Zheng et al., 2025) adopts a sequence-level reward formulation, in contrast to the token-level objective used in GRPO, which improves training stability for LLMs, especially in mixture of experts settings. Alternative ap-

proaches demonstrate that reinforce style policy gradient methods can attain competitive performance and stable training when appropriately tuned (Chu et al., 2025; Ahmadian et al., 2024). For example, REINFORCE++ (Hu, 2025) introduces global advantage normalization to improve the training stability of vanilla policy gradient methods. ReMax (Li et al., 2023) modifies the reinforce gradient estimator by introducing a subtractive baseline to reduce variance and improve training stability. Specifically, the baseline is computed by greedily sampling a response and evaluating its corresponding reward.

**Diversity in Reinforcement Learning.**  Exploration is essential for performance improvement in high dimensional reinforcement learning (Ladosz et al., 2022). Previous studies (Yao et al., 2025; Cheng et al., 2025; Wang et al., 2025) primarily adopt entropy-based regularization to balance exploitation and exploration. These approaches treat entropy as a proxy for diversity by maximizing the average output entropy of LLMs, rewarding trajectories with higher token-level uncertainty to encourage more diverse generations. (Masood & Doshi-Velez, 2019) instead propose to exploit maximum mean discrepancy distance as a trajectory level regularizer to enhance diversity of the learned policy. Recent studies (Lanchantin et al., 2025; Chung et al., 2025; Ismayilzada et al., 2025) explicitly incorporate both quality and diversity into preference optimization.

Concurrent work (Chen et al., 2025a; Li et al., 2025) also explores trajectory-level semantic diversity measures for RL fine-tuning of LLMs. While these methods similarly rely on trajectory-level semantic similarity to quantify diversity, SetPO adopts a fundamentally different formulation. We introduce a set-level objective that evaluates the generated batch as a holistic distribution. By employing a leave-one-out marginal calculation, we explicitly measure the contribution of each trajectory to the group's total information mass, thereby optimizing diversity from a global distributional perspective rather than through pairwise separation explored in previous studies.

# 3. Diversity-Aware Reward Calculation

## 3.1. Preliminaries: Group-based Policy Optimization

Group-based policy optimization has emerged as a stable and efficient alternative to standard PPO for language model alignment, primarily by eliminating the need for a separate critic network. One of the representative methods in this family is GRPO.

Formally, for each prompt $x$ sampled from the dataset $\mathcal{D}$, GRPO generates a group of $G$ outputs $\{o_1, \ldots, o_G\}$ from the current policy $\pi_{\theta_{\mathrm{old}}}$. Denote $\rho_{i,t}(\theta) = \frac{\pi_\theta(a_{i,t}|x,a_{i,<t})}{\pi_{\theta_{\mathrm{old}}}(a_{i,t}|x,a_{i,<t})}$ is the importance sampling ratio, and $\mathbb{D}_{\mathrm{KL}}$ is the KL diver-

gence penalty to keep the policy close to a reference model $\pi_{\mathrm{ref}}$. The reward is assigned with a scalar score $r_i$ to each trajectory $o_i$ by the model. To reduce variance without a learned value function, GRPO computes the advantage $\bar{A}_i$ for each output by normalizing the rewards relative to the group statistics: $\bar{A}_i = \frac{r_i - \mu(\{r_j\}_{j=1}^G)}{\sigma(\{r_j\}_{j=1}^G) + \epsilon}$, where $\mu$ and $\sigma$ denote the mean and standard deviation of the rewards within the group. The policy parameters $\theta$ are updated by maximizing the following surrogate objective:

$$
\begin{aligned}
J(\theta) =\mathbb{E}\Bigg[ &\frac{1}{G} \sum_{i=1}^G \frac{1}{|o_i|} \sum_{t=1}^{|o_i|} \min\Big(\rho_{i,t}(\theta)\, \bar{A}_i, \\
&\mathrm{clip}(\rho_{i,t}(\theta), 1-\varepsilon, 1+\varepsilon)\, \bar{A}_i\Big) - \\
&\beta\, \mathrm{KL}\big(\pi_\theta(\cdot \mid x, a_{i,<t}) \,\big\|\, \pi_{\mathrm{ref}}(\cdot \mid x, a_{i,<t})\big) \Bigg].
\end{aligned}
\tag{1}
$$

## 3.2. Diversity of a Language Model

We first study the diversity measure of a language model at the level of trajectories. Given a prompt $x$, a trajectory is defined as a finite token sequence $y = (a_1, \ldots, a_T) \in \Omega$, where $a_T$ is the *EOS* token or $T$ reaches the maximum length. Accordingly, a parameterized language model induces a trajectory distribution following $P_\theta(y|x) = \prod_{t=1}^T \pi_\theta(a_t|x, a_{<t})$. In the following discussion, we omit the subscript $\theta$ where unambiguous.

Unlike methods that focus on diversity at the token level, we evaluate complete outputs semantically. We utilize a bounded, symmetric similarity kernel $k : \Omega \times \Omega \to [0,1]$, satisfying $k(y, y') = k(y', y)$ and $k(y, y) = 1$. Specifically, we let $k(y, y') = \mathrm{sim}(e(y), e(y'))$, employing a pre-defined semantic embedding function $e(\cdot) : \Omega \to \mathbb{R}^n$, where $n$ denotes the dimension of the embedding space.

For any trajectory distribution $P \in \mathcal{P}(\Omega)$, we define the kernelized local mass of a single trajectory $y$ as:

$$
m_P(y) := \mathbb{E}_{y' \sim P}\left[k(y, y')\right].
$$

Conceptually, $m_P(y)$ estimates the semantic density surrounding $y$. A high value indicates that $y$ resides in a crowded neighborhood, implying redundancy. Conversely, a low value places $y$ in a sparse region, signifying relative novelty. We then quantify the language model's overall diversity by aggregating these local estimates. Let $g : [0,1] \to \mathbb{R}$ be a continuous, non-increasing function. The diversity objective is defined as:

$$
\mathcal{F}(P) := \mathbb{E}_{y \sim P}\Big[g\big(m_P(y)\big)\Big].
\tag{2}
$$

Since $g$ is non-increasing, a smaller local mass $m_P(y)$ yields a larger return. This mechanism effectively rewards

semantic novelty. In this paper, we adopt the following shaping function: $g(x) = -\log(1 + x)$. Beyond numerical stability, this formulation embodies the principle of diminishing marginal sensitivity. The gradient magnitude is steepest when $x \approx 0$ and decays as $x$ increases. Consequently, the function prioritizes the preservation of highly unique trajectories, as the reward for adjusting already crowded regions diminishes.

### 3.3. Leave-one-out Margin for a Single Trajectory

We now derive a per-sample marginal assignment by from the aspect of assessing the diversity of a language model through Monte Carlo approximations over arbitrary subsets. Let $S \subset \Omega$ be any finite collection of i.i.d. trajectories drawn from $P_\theta(\cdot \mid x)$ with cardinality $|S| \geq 3$. Since $S$ serves as a Monte Carlo sample of the trajectory space, we can estimate the population diversity $\mathcal{F}(P)$ using a generalized within-group score $D(S)$:

$$D(S) := \frac{1}{|S|} \sum_{y \in S} g\big(\widehat{m}(y; S)\big), \qquad (3)$$

where $\widehat{m}(y; S) := \frac{1}{|S|-1} \sum_{z \in S \setminus \{y\}} k(y, z)$. Here, $\widehat{m}(y; S)$ estimates the local semantic density around $y$ based on the empirical distribution of $S$. Since the samples are i.i.d., $D(S)$ remains a consistent estimator for any valid subset $S$.

This statistical property allows us to quantify the contribution of a single trajectory $o_i$ by contrasting the diversity estimate of the full generated batch $\Omega = \{o_1, \ldots, o_G\}$ against that of the subset $\Omega \setminus \{o_i\}$. If $o_i$ is informative, its presence should improve the resolution of the approximation compared to the subset that excludes it. We therefore define the diversity contribution $s_i$ of each trajectory $o_i$ towards the considered whole set as the leave-one-out marginal difference given as:

$$s_i := D(\Omega) - D\big(\Omega \setminus \{o_i\}\big). \qquad (4)$$

In this formulation, $D(\Omega)$ utilizes all $G$ samples to estimate the kernelized density, while $D(\Omega \setminus \{o_i\})$ re-evaluates the score using the remaining $G - 1$ trajectories. Explicitly, the term for the reduced set is computed as:

$$D(\Omega \setminus \{o_i\}) = \frac{1}{G-1} \sum_{j \neq i} g\big(\widehat{m}(o_j; \Omega \setminus \{o_i\})\big).$$

A larger $s_i$ indicates that including $o_i$ yields a more significant gain in the estimated diversity of the set, implying that $o_i$ captures a distinct semantic mode not covered by other samples. Conversely, a small or negative $s_i$ suggests redundancy. Since the group size $G$ in reinforcement learning is typically small, computing these recursive leave-one-out marginals imposes a negligible computational burden.

### 3.4. Set-Level Policy Optimization Methods

We integrate the marginal diversity contribution for each sample towards a certain set derived above as a plug-in shaping term for the group policy optimization framework introduced in the Preliminaries. Our method requires only the standard group of sampled trajectories $\{o_i\}_{i=1}^{G}$ and the base advantages $\bar{A}_i$.

We compute the diversity score $s_i$ from the same group and form the augmented advantage:

$$\hat{A}_i := \bar{A}_i + \lambda\, s_i,$$

where $\lambda \geq 0$ controls the trade-off between task performance and diversity. We then substitute this augmented advantage directly into the standard GRPO objective defined in Eq. (1), replacing the original base advantage. The diversity plug-in solely modifies the advantage term $\bar{A}_i$. This modular design ensures that our diversity-aware reward calculation is compatible with a broad family of set-level RL objectives including DAPO, GSPO and so on, provided they rely on a group-based advantage estimation.

## 4. Theoretical Analysis

### 4.1. Population view

To understand the change of the diversity $\mathcal{F}(P)$ towards any single trajectory, we take a perturbation perspective. Firstly, we give the assumptions on the kernel and the shaping functions as follows.

**Assumption 4.1.** The kernel function $k(\cdot, \cdot)$ is measurable and symmetric, i.e. $k(x, y) = k(y, x)$. Besides, it holds that $0 \leq k(x, y) \leq 1$ for all $x, y$. The shaping function $g : [0, 1] \to \mathbb{R}$ is continuously differentiable on $[0, 1]$ and $g'$ is bounded. Besides $g$ is non-increasing on $[0, 1]$.

Consider the mixture update $P_\varepsilon = (1 - \varepsilon)P + \varepsilon \delta_\tau$, which can be seen as a perturbation of the original distribution by single trajectory $\tau$. We characterize the influence of $\tau$ through Gâteaux derivative.

**Theorem 4.2** (Influence function of the diversity functional). *Under Assumption 4.1, the Gâteaux derivative of $\mathcal{F}$ at $P$ in the direction $\tau$ is*

$$\mathcal{I}(\tau; P) := \frac{d}{d\varepsilon} \mathcal{F}(P_\varepsilon)\Big|_{\varepsilon=0} = \underbrace{g(m_P(\tau))}_{\textit{intrinsic novelty}}$$
$$+ \underbrace{\mathbb{E}_{y \sim P}[g'(m_P(y))\, k(y, \tau)]}_{\textit{interaction penalty}} - \Psi(P),$$

*where $\Psi(P)$ does not depend on $\tau$.*

The concrete technical derivations are omitted here and we defer all proofs to Appendix C. The influence splits into two

*Table 1.* Pass@1 accuracy on mathematical benchmarks for Qwen2.5-Math-7B. Entries marked with * are from the original paper.

| Method | GSM8K | MATH500 | College Math | AMC23 | AIME24 | AIME25 | Avg |
|---|---|---|---|---|---|---|---|
| Qwen2.5-Math-7B | 53.4 | 48.1 | 19.4 | 41.3 | 9.9 | 4.0 | 29.4 |
| R1-zero-Div* | 90.6 | 76.9 | 47.5 | - | - | - | - |
| GRPO | 89.1 | 73.5 | 42.2 | 53.5 | 15.4 | 9.7 | 47.2 |
| GSPO | 89.5 | 73.0 | 45.0 | 57.7 | 17.4 | 7.9 | 48.4 |
| DAPO | 92.2 | 77.6 | 47.1 | 59.6 | 21.7 | 11.0 | 51.7 |
| **SetPO**+GRPO | 92.2 (+3.1) | **80.8** (+7.3) | 48.3 (+6.1) | 60.5 (+7.0) | 21.6 (+8.2) | **13.6** (+3.9) | 52.8 (+5.6) |
| **SetPO**+GSPO | 91.6 (+2.1) | 75.7 (+2.7) | 47.2 (+2.2) | 60.9 (+3.2) | 21.1 (+3.7) | 10.1 (+2.2) | 51.1 (+2.7) |
| **SetPO**+DAPO | **93.0** (+0.8) | 79.2 (+1.6) | **49.5** (+2.2) | **62.3** (+2.7) | **25.3** (+3.6) | 13.5 (+2.5) | **53.8** (+2.1) |

complementary mechanisms: (i) an *intrinsic novelty* term $g(m_P(\tau))$ that strictly favors low local mass since $g$ is non-increasing; and (ii) an *interaction penalty* that discourages trajectories that are highly similar to what already exists, since $g'(m_P(y)) < 0$ and the kernel weight $k(y, \tau)$ concentrates the penalty on nearby regions. Thus, the $\mathcal{F}(P)$ follows diminishing-returns principles and is anti-redundant.

### 4.2. Leave-one-out as marginal contribution

Our algorithm operates on a finite candidate set and repeatedly removes the element that hurts diversity the least. We need to stress that the leave-one-out (LOO) score (4) is not a heuristic proxy. It is exactly the marginal decrease in the objective induced by removing $o_i$. The following results make this precise and show that $s_i$ systematically penalizes redundancy. For finite set $\Omega$, we characterize the marginal contribution of $o_i$ as follows.

**Theorem 4.3** (Exact decomposition of the LOO contribution). *For $|\Omega| \geq 3$, $s_i$ admits an exact decomposition into three parts:*

$$s_i = \underbrace{\frac{1}{|\Omega|}g(m_i)}_{\mathcal{T}_{self}} + \underbrace{\left(\frac{1}{|\Omega|} - \frac{1}{|\Omega|-1}\right)\sum_{j\neq i}g(m_j)}_{\mathcal{T}_{norm}}$$
$$+ \underbrace{\frac{1}{|\Omega|-1}\sum_{j\neq i}\left[g(m_j) - g(m_j^{(-i)})\right]}_{\mathcal{T}_{interaction}},$$

*where $m_j^{(-i)} = \widehat{m}(o_j; \Omega \setminus \{o_i\})$.*

The concrete technical derivations are omitted here and we defer all proofs and a complete analysis to Appendix D. Moreover, each interaction summand depends on $o_i$ only through the pairwise similarity $k(o_j, o_i)$.

The decomposition illustrates why $s_i$ serves as a robust signal. The first term, $\mathcal{T}_{\text{self}}$, rewards points that are individually novel under the diversity shaping function $g$. Meanwhile,

the interaction term, $\mathcal{T}_{\text{interaction}}$, measures the extent to which $o_i$ congests the neighborhood structure. Removing a redundant point typically reduces the local mass of its neighbors, thereby increasing their $g(\cdot)$ values. This raises $D(\Omega \setminus \{o_i\})$ and consequently yields a smaller $s_i$. This mechanism captures exactly the diminishing-returns behavior desired for the leave-one-out construction.

Furthermore, we consider our marginal contribution towards each similarity between current sample and others.

**Theorem 4.4** (Strict anti-redundancy of $s_i$). *For any $t \neq i$, if Assumption 4.1 holds, the leave-one-out contribution is strictly decreasing in the pairwise similarity $k(o_i, o_t)$:*

$$\frac{\partial s_i}{\partial k(o_i, o_t)} = \frac{1}{|\Omega|(|\Omega|-1)}\Big(g'(m_i) + g'(m_t)\Big) < 0.$$

Increasing similarity to any other element must reduce the marginal contribution of $o_i$ under our objective. Therefore, iterative removal of the smallest-$s_i$ elements is provably a redundancy-removal procedure. Finally, we establish a global ordering guarantee. If one point is uniformly less similar to the rest of the set, then leave-one-out necessarily ranks it higher.

**Theorem 4.5** (Dominance ordering). *Let $o_r, o_c \in \Omega$. If Assumption 4.1 holds and $k(o_r, o_j) \leq k(o_c, o_j)$ for all $j \notin \{r, c\}$, and strict inequality holds for at least one such $j$, then it holds that $s_r > s_c$.*

In summary, Theorem 4.2 quantifies how injecting a single trajectory perturbs the population-level diversity functional through its influence. A trajectory is favored when it lies in low-mass regions, and it is penalized according to its similarity-induced overlap with the existing distribution. On a finite candidate set, Theorems 4.3–4.5 further establish that the leave-one-out margin $s_i$, defined as the exact marginal decrease of the same objective when removing $o_i$, is strictly decreasing in every pairwise similarity $k(o_i, o_t)$. Therefore, $s_i$ provides a principled, anti-redundant ranking signal whose optimization is directly aligned with increasing

*Table 2.* Pass@1 accuracy on mathematical benchmarks for Qwen2.5-Math-1.5B. Entries marked with * are from the original paper.

| Method | GSM8K | MATH500 | College Math | AMC23 | AIME24 | AIME25 | Avg |
|---|---|---|---|---|---|---|---|
| Qwen2.5-Math-1.5B | 39.4 | 36.4 | 6.6 | 29.5 | 4.6 | 1.8 | 19.7 |
| R1-zero-Div* | 83.2 | 70.4 | 43.9 | - | - | - | - |
| GRPO | 83.0 | 67.2 | 43.1 | 47.5 | 11.5 | 8.2 | 43.4 |
| GSPO | 83.5 | 67.2 | 43.0 | 48.8 | 11.1 | 5.6 | 43.2 |
| DAPO | 85.9 | 69.5 | 44.3 | 50.1 | 12.0 | 6.6 | 44.7 |
| **SetPO**+GRPO | **86.9** (+3.9) | **72.9** (+5.7) | 45.7 (+2.6) | 51.3 (+3.8) | **13.4** (+1.9) | **9.7** (+1.5) | **46.7** (+3.3) |
| **SetPO**+GSPO | 85.0 (+1.5) | 69.4 (+2.2) | 44.7 (+1.7) | 50.6 (+1.8) | 13.2 (+2.1) | 7.4 (+1.8) | 45.2 (+2.0) |
| **SetPO**+DAPO | 86.6 (+0.7) | 71.7 (+2.2) | **46.5** (+2.2) | **53.2** (+3.1) | **13.4** (+1.4) | 8.9 (+2.3) | **46.7** (+2.0) |

the diversity objective under the induced empirical distribution, rather than serving as a heuristic.

# 5. Numerical Experiments

## 5.1. General Mathematical Reasoning

**Backbone Models.** We employ two instruction-tuned backbone models of varying scales to verify the generalizability of our approach: Qwen2.5-Math-7B, Qwen2.5-Math-1.5B (Yang et al., 2024). Our implementation is built upon the open-source veRL framework (Sheng et al., 2024) to ensure efficient training and flexible rollout management.

**Baselines.** We benchmark our proposed method SetPO+GRPO, SetPO+GSPO and SetPO+DAPO against four representative reinforcement learning baselines: GRPO (Shao et al., 2024), GSPO (Zheng et al., 2025), DAPO (Yu et al., 2025) and R1-Zero-Div (Yao et al., 2025). To ensure a rigorous and fair comparison, we strictly unify the experimental protocols across all methods; specifically, we utilize identical hyperparameters including learning rate, batch size, and total training steps.

**Training Configuration.** All models are trained using the standard GSM8K training split (Cobbe et al., 2021). We optimize the models for 2 epochs using the AdamW optimizer with a constant learning rate of $3 \times 10^{-6}$ and a global batch size of 64. During the online data generation phase, we set the rollout number to 6 per prompt and utilize a sampling temperature of 0.9 to encourage diverse exploration in the solution space. More concrete experiment settings can be found in Appendix E.

**Evaluation Configuration.** We assess performance across a comprehensive suite of mathematical reasoning benchmarks, including GSM8K test, MATH500 (Hendrycks et al., 2021), Olympiad (He et al., 2024), college math bench (Tang et al., 2024), AMC 2023 (Mathematical Asso-

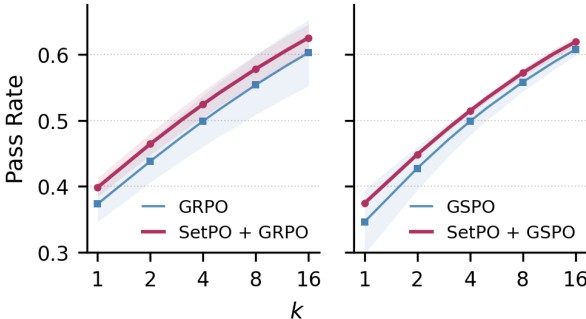

*Figure 2.* Performance of SetPO+GRPO and SetPO+GSPO on the Olympiad benchmark. Shaded regions indicate variance across runs. Both methods improve performance and reduce variability, suggesting that SetPO stabilizes the training dynamics.

ciation of America, 2023), and AIME 24/25 (Mathematical Association of America, 2025). To ensure statistical robustness, all reported results are averaged over five random seeds. Unless otherwise specified, we report Pass@1 accuracy derived from Avg@64 (i.e., averaging over 64 sampled generations) with an inference-time temperature of 0.5. For the college math bench, we adopt Avg@16 for efficiency due to the significantly larger size of its evaluation set.

**Numerical Results.** Tables 1 and 2 summarize the Pass@1 results for the 7B and 1.5B models, respectively. Overall, SetPO achieves large margin gains over baseline methods across model scales and benchmark difficulties. In particular, SetPO consistently improves GRPO, GSPO, and DAPO across benchmarks and model scales. As shown in Table 1, SetPO+DAPO achieves the best overall performance with an average Pass@1 accuracy of 53.8%. Notably, the most pronounced improvement is observed when applying SetPO to GRPO, which increases the average accuracy from 47.2% to 52.8% (+5.6). On challenging benchmarks, SetPO yields strong relative gains; for example, on AIME24, accuracy improves from 15.7% to 21.6%, corresponding to a

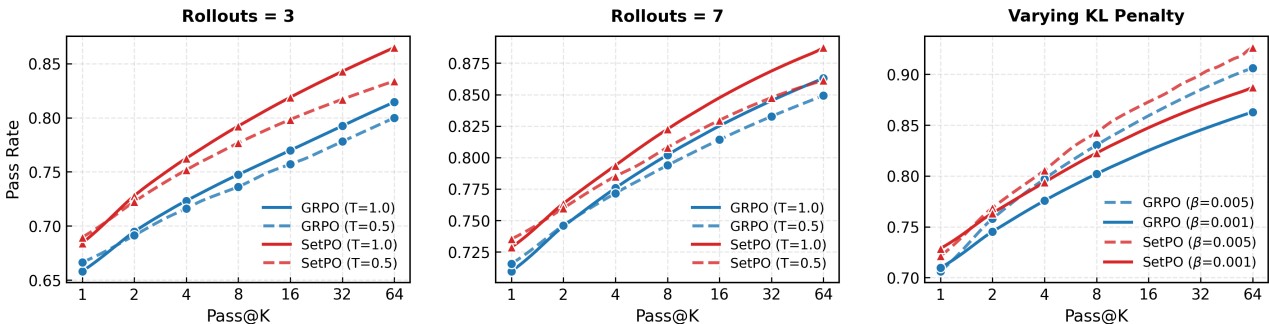

*Figure 3.* Pass@K performance on the Countdown task for SetPO (SetPO+GRPO) versus the GRPO baseline under varying decoding temperatures, training rollout counts, and KL penalties. SetPO consistently achieves higher Pass@K across these settings.

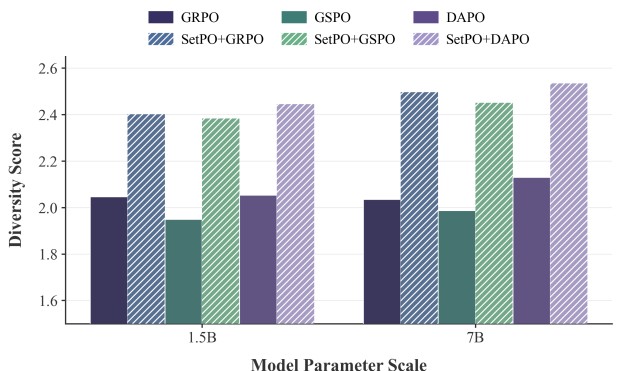

*Figure 4.* Diversity scores across the AIME24 benchmark, computed per problem from 16 Gemini-2.5-Flash generated solutions. Scores range from 1 (lowest diversity) to 5 (highest diversity).

relative increase of over 40%. Table 2 further demonstrates a similar trend on the 1.5B models, where SetPO+DAPO achieves the best overall average accuracy of 46.7%. In addition, as shown in Figure 2, our method significantly reduces the variance of the baselines and yields consistent performance gains across different random seeds.

**Diversity Assessment.** To assess the diversity of the generated answers, we utilize Gemini-2.5-Flash as an LLM evaluator. The scoring adopts a 5-point scale, ranging from minimum diversity 1 to maximum diversity 5. The full evaluation prompt is provided in Appendix H. Figure 4 shows that SetPO-enhanced methods consistently achieve higher diversity scores than the baselines across both 1.5B and 7B model scales, demonstrating that our algorithm effectively improves the diversity of final outputs.

### 5.2. Countdown Evaluation

**Task Description.** The Countdown task is a rigorous benchmark for arithmetic reasoning and constraint satisfaction. In this task, the model is provided with a target number and a set of input numbers. The goal is to generate

a valid mathematical expression using the input numbers and basic arithmetic operations that evaluates to the target. This task challenges the model's ability to perform search in a discrete action space and verify its own reasoning steps.

**Model and Training.** We conduct experiments on the countdown dataset (Pang et al., 2023) using Qwen2.5-3B as the backbone model. During training, we optimize all models with a global batch size of 128 and a maximum generation length of 2048. To enforce strict adherence to the required answer format, our rule-based reward function incorporates an auxiliary format reward term with a weight of 0.1. We explore varied hyperparameter settings, specifically considering rollout numbers in $\{3, 7\}$ and KL divergence coefficients of $\{0.001, 0.005\}$. Detailed configurations are provided in Appendix E.

**Evaluation Protocol.** During inference, we report the unbiased Pass@k metric for $k \in \{1, \ldots, 64\}$. To provide a comprehensive analysis of the model's behavior, we investigate the impact of sampling temperature by evaluating performance at $T = 0.5$ and $T = 1.0$, and discuss how sample size variations influence empirical conclusions.

**Performance Superiority.** As illustrated in Figure 3, SetPO consistently outperforms the GRPO baseline on the Countdown task, demonstrating superior sample efficiency and solution quality. This performance advantage remains consistent under both 3 rollout and 7 rollout settings. Notably, the performance gap widens as $k$ increases, indicating that SetPO is particularly effective at covering more rewarding modes compared to the baseline. Furthermore, our method maintains a distinct advantage across diverse sampling configurations. While a high temperature generally yields improved pass rates for both methods by encouraging broader exploration, SetPO consistently surpasses GRPO at temperatures of both 0.5 and 1.0. Finally, the analysis of varying KL penalties shown in the right panel of Figure 3 further validates the stability of SetPO, showing that it

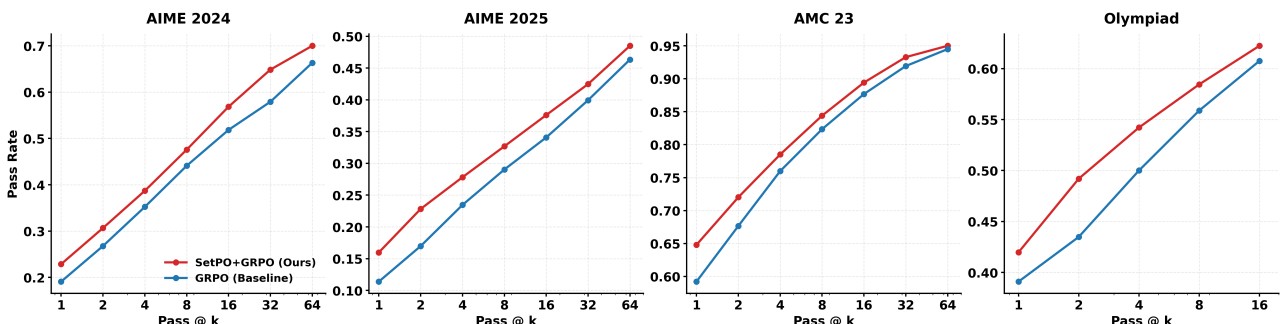

*Figure 5.* Pass@k performance of SetPO+GRPO versus the GRPO baseline on Qwen2.5-32B. SetPO+GRPO consistently outperforms GRPO across all k, demonstrating robust gains under varying sampling budgets.

*Table 3.* Diversity statistics for the Countdown task. Diversity width measures the count of problems yielding at least two different outputs. Average mode represents the mean number of distinct answers per question. Sampling temperature is set to 1.0.

| Setting | Diversity Width | Average Mode |
|---|---|---|
| Rollout 7 SetPO+GRPO | **415** | **3.5** |
| Rollout 7 GRPO | 351 | 2.3 |
| Rollout 3 SetPO+GRPO | **403** | **3.2** |
| Rollout 3 GRPO | 321 | 2.1 |

yields robust performance improvements over the baseline irrespective of the regularization strength.

**Diversity Analysis.** To systematically evaluate the output diversity, we analyze the statistics presented in Table 3. We introduce two primary metrics: (1) diversity width, defined as the count of samples for which the model generates at least two distinct correct solutions; and (2) average mode, which calculates the average number of unique valid answers specifically for those instances identified diverse.

The empirical results demonstrate that integrating SetPO significantly expands the solution space compared to the GRPO baseline. In the 7-rollout setting, SetPO+GRPO increases the diversity width from 351 to 415, indicating a robust capability to discover alternative reasoning paths. Furthermore, the average mode rises from 2.3 to 3.5. This quantitative improvement confirms that SetPO+GRPO actively promotes a broader coverage of valid solutions.

### 5.3. Scaling Up to 32B Models

**Experimental Settings.** We further validate the scalability of our approach by extending the experiments to the 32B parameter scale, utilizing Qwen2.5-32B (Team et al., 2024) as the backbone model. The models are fine-tuned on the challenging hard subset of Simple-RL Zoo (Zeng et al., 2025). To ensure stable optimization and effective

exploration at this scale, we configure the global batch size to 1024 and the PPO mini-batch size to 256. During the inference rollout phase, we generate 6 trajectories per prompt with a sampling temperature of 0.1 to maintain a balance between diversity and precision.

**Numerical Performance.** Figure 5 verifies the scalability of SetPO on 32B-parameter models, where it consistently outperforms the GRPO baseline across all four mathematical benchmarks. Specifically, SetPO shows consistent advantages in both single-sample and multi-sample regimes. In the Pass@1 setting, our method achieves a clear accuracy improvement over the baseline, especially on the more challenging AIME 2025 and Olympiad tasks, suggesting stronger reasoning quality. On hard benchmarks such as AIME24 and AIME25, SetPO provides an almost 5 percentage-point gain in Pass@1 compared to vanilla GRPO. Moreover, as the sampling budget $k$ increases, SetPO consistently preserves this performance gap. These results indicate that SetPO not only improves the correctness of individual solutions, but also enhances the diversity and robustness of the generation distribution for complex reasoning tasks.

## 6. Conclusion

We introduced SetPO, a set-level policy optimization method that promotes output diversity by rewarding trajectories according to their leave-one-out marginal contribution within the rollout set. SetPO estimates a local kernel density induced by semantic trajectory similarity and converts each trajectory's density-sensitive marginal effect into an interpretable leave-one-out diversity credit, which can be incorporated into standard group-based policy optimization. We analyze the diversity functional under distributional perturbations, establishing bounds and a monotonic mechanism showing that rarer modes yield larger diversity gains. Extensive experiments across a wide range of model sizes on mathematical reasoning and combinatorial tasks show that SetPO consistently outperforms strong baselines, achieving higher accuracy and maintaining solution diversity.

## Impact Statement

This paper presents work whose goal is to advance the field of reinforcement learning. There are many potential societal consequences of our work, none which we feel must be specifically highlighted here.

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

## A. Brief Introduction to DAPO and GSPO

**DAPO.** Decoupled Clip and Dynamic sAmpling Policy Optimization (DAPO) is a GRPO-style RL algorithm tailored to verifiable reasoning tasks, where for each prompt $q$ one samples a group of responses $\{o_i\}_{i=1}^{G}$ from the behavior policy $\pi_{\theta_{\text{old}}}$ and optimizes a clipped surrogate objective with *asymmetric* clipping ranges to mitigate entropy collapse ("Clip-Higher"). Concretely, DAPO maximizes

$$J_{\text{DAPO}}(\theta) = \mathbb{E}_{(q,a)\sim\mathcal{D},\,\{o_i\}_{i=1}^{G}\sim\pi_{\theta_{\text{old}}}(\cdot|q)}\left[\frac{1}{\sum_{i=1}^{G}|o_i|}\sum_{i=1}^{G}\sum_{t=1}^{|o_i|}\min\left(r_{i,t}(\theta)\,\bar{A}_{i,t},\,\text{clip}(r_{i,t}(\theta),\,1-\epsilon_{\text{low}},\,1+\epsilon_{\text{high}})\,\bar{A}_{i,t}\right)\right],$$

subject to the dynamic-sampling constraint

$$0 < \left|\left\{\,o_i : \text{is\_equivalent}(a,o_i)\,\right\}\right| < G.$$

Here $r_{i,t}(\theta) = \frac{\pi_\theta(o_{i,t}|q,o_{i,<t})}{\pi_{\theta_{\text{old}}}(o_{i,t}|q,o_{i,<t})}$ is the token-level importance ratio, and $\hat{A}_{i,t}$ is a group-normalized advantage (shared across tokens within the same response) computed from outcome rewards $\{R_i\}_{i=1}^{G}$ via $\hat{A}_{i,t} = \frac{R_i - \text{mean}(\{R_i\})}{\text{std}(\{R_i\})}$.

**GSPO.** Group Sequence Policy Optimization (GSPO) addresses a mismatch in GRPO between token-level importance ratios and sequence-level rewards by defining the importance ratio at the *sequence* level and applying *sequence-level* clipping. GSPO maximizes

$$J_{\text{GSPO}}(\theta) = \mathbb{E}_{x\sim\mathcal{D},\,\{y_i\}_{i=1}^{G}\sim\pi_{\theta_{\text{old}}}(\cdot|x)}\left[\frac{1}{G}\sum_{i=1}^{G}\min\left(s_i(\theta)\,\bar{A}_i,\,\text{clip}(s_i(\theta),\,1-\epsilon,\,1+\epsilon)\,\bar{A}_i\right)\right],$$

where the group-based advantage is $\bar{A}_i = \frac{r(x,y_i)-\text{mean}(\{r(x,y_i)\})}{\text{std}(\{r(x,y_i)\})}$ and the (length-normalized) sequence-level importance ratio is

$$s_i(\theta) = \left(\frac{\pi_\theta(y_i\mid x)}{\pi_{\theta_{\text{old}}}(y_i\mid x)}\right)^{\frac{1}{|y_i|}} = \exp\left(\frac{1}{|y_i|}\sum_{t=1}^{|y_i|}\log\frac{\pi_\theta(y_{i,t}\mid x,y_{i,<t})}{\pi_{\theta_{\text{old}}}(y_{i,t}\mid x,y_{i,<t})}\right).$$

This design aligns the optimization unit (sequence) with the reward granularity, reducing variance and improving stability, especially for long responses and large-scale training.

## B. Our Algorithm

The target function used for SetPO+GRPO is given as

$$\begin{aligned}J(\theta) =&\mathbb{E}\left[\frac{1}{G}\sum_{i=1}^{G}\frac{1}{|o_i|}\sum_{t=1}^{|o_i|}\min\left(\rho_{i,t}(\theta)\,\hat{A}_i,\,\text{clip}(\rho_{i,t}(\theta),1-\varepsilon,1+\varepsilon)\,\hat{A}_i\right)-\right.\\ &\left.\beta\,\text{KL}\big(\pi_\theta(\cdot\mid x,a_{i,<t})\,\big\|\,\pi_{\text{ref}}(\cdot\mid x,a_{i,<t})\big)\right],\end{aligned} \tag{5}$$

where $\hat{A}_i$ combines the original advantage $\bar{A}_i$ with our leave-one-out marginal, calculated by (4). The complete algorithm of our SetPO+GPRO is given in Algorithm 1.

## C. Theoretical Framework of $\mathcal{F}(P)$: Trajectory Diversity

In this section, we establish the mathematical foundations of our diversity measure. We begin by defining the diversity measure for large language models, derive its sensitivity to perturbations, establish a rigorous lower bound for finite-step updates, and finally discuss its thermodynamic properties in relation to decoding temperature.

---

**Algorithm 1** Set-Level Policy Optimization

---

**Require:** Prompt dataset $\mathcal{D}$, Policy $\pi_\theta$, Reference model $\pi_{\text{ref}}$, Group size $G$, Diversity coefficient $\lambda$, KL coefficient $\beta$, Learning rate $\eta$.

1: **while** not reach max iteration **do**
2:      Sample a batch of prompts $\Omega = \{x_1, \ldots, x_M\}$ from $\mathcal{D}$.
3:      **for** each prompt $x \in \Omega$ **do**
4:          **Sampling:**
5:          Generate $G$ outputs $A = \{o_1, \ldots, o_G\}$ where $o_i \sim \pi_{\theta_{\text{old}}}(\cdot \mid x)$.
6:          **Reward Calculation:**
7:          Compute task rewards $\{r_1, \ldots, r_G\}$ for each output.
8:          Compute base advantages $\bar{A}_i$ (e.g., via group normalization: $\bar{A}_i = \frac{r_i - \text{mean}(r)}{\text{std}(r) + \epsilon}$).
9:          **Diversity Estimation (Leave-one-out):**
10:         Compute pairwise similarity matrix $K \in \mathbb{R}^{G \times G}$ where $K_{ij} = k(o_i, o_j)$.
11:         Calculate group score $D(A) = \frac{1}{G} \sum_{i=1}^{G} g(\widehat{m}_i)$.
12:         **for** $i = 1$ to $G$ **do**
13:            Calculate reduced-set score $D(\Omega \setminus \{o_i\})$ using Eq. 3 (recomputing local masses).
14:            Compute diversity contribution $s_i = D(\Omega) - D(\Omega \setminus \{o_i\})$.
15:         **end for**
16:         Compute augmented advantage: $\hat{A}_i \leftarrow \bar{A}_i + \lambda \cdot s_i$.
17:      **end for**
18:      **Policy Optimization:**
19:      Maximize the surrogate objective $J(\theta)$ by (5) on the collected batch as $\theta \leftarrow \theta + \eta \nabla_\theta J(\theta)$.
20: **end while**

---

## C.1. Preliminaries and Definitions

Let $\Omega$ be the trajectory space. We employ a bounded, symmetric similarity kernel $k : \Omega \times \Omega \to [0, 1]$ with $k(y, y) = 1$. For any distribution $P \in \mathcal{P}(\Omega)$, the *kernelized local mass* at $y$ is defined as:

$$m_P(y) := \mathbb{E}_{y' \sim P}[k(y, y')] \in [0, 1]. \tag{6}$$

This quantity estimates the semantic density of the distribution $P$ around trajectory $y$. The global diversity functional is defined as:

$$\mathcal{F}(P) := \mathbb{E}_{y \sim P}\Big[g\big(m_P(y)\big)\Big], \tag{7}$$

where the shape function $g(x) : [0, 1] \to \mathbb{R}$. A canonical choice used in this work is $g(x) = -\log(1 + x)$, which is smooth and strictly decreasing.

**Assumption C.1.** The kernel function $k(\cdot, \cdot)$ is measurable and symmetric, i.e. $k(x, y) = k(y, x)$. Besides, it holds that $0 \le k(x, y) \le 1$ for all $x, y$. The shape function $g(x) : [0, 1] \to \mathbb{R}$ is a continuous, strictly decreasing shaping function and there exists $L < \infty$ such that $\|g'(a) - g'(b)\| \le L\|a - b\|$ for all $a, b \in [0, 1]$.

For a certain trajectory $\tau \in \Omega$ and consider the mixture perturbation

$$P_\varepsilon := (1 - \varepsilon)P + \varepsilon \, \delta_\tau, \qquad \varepsilon \in [0, 1]. \tag{8}$$

We study the finite-$\varepsilon$ change $\mathcal{F}(P_\varepsilon) - \mathcal{F}(P)$ and its relation to the original rarity $m_P(\tau)$. For any $y \in \Omega$, it holds that

$$m_{P_\varepsilon}(y) = \mathbb{E}_{y' \sim P_\varepsilon} k(y, y') = (1 - \varepsilon)\mathbb{E}_{y' \sim P} k(y, y') + \varepsilon \mathbb{E}_{y' \sim \delta_\tau} k(y, y')$$
$$= (1 - \varepsilon)m_P(y) + \varepsilon k(y, \tau) = m_P(y) + \varepsilon\Big(k(y, \tau) - m_P(y)\Big). \tag{9}$$

Define the *similarity offset*

$$d_\tau(y) := k(y, \tau) - m_P(y).$$

Then (9) becomes

$$m_{P_\varepsilon}(y) = m_P(y) + \varepsilon \, d_\tau(y). \tag{10}$$

In particular, using $k(\tau, \tau) = 1$, it yields

$$m_{P_\varepsilon}(\tau) = (1 - \varepsilon)m_P(\tau) + \varepsilon = m_P(\tau) + \varepsilon\big(1 - m_P(\tau)\big).$$

It is immediate from this expression that rarer samples (smaller $m_P(\tau)$) receive a larger injection boost, since the increment $m_{P_\varepsilon}(\tau) - m_P(\tau) = \varepsilon\big(1 - m_P(\tau)\big)$ increases as $m_P(\tau)$ decreases.

We next perform decomposition of the finite-$\varepsilon$ diversity change of $\mathcal{F}(P)$. By definition,

$$\mathcal{F}(P_\varepsilon) = \mathbb{E}_{y \sim P_\varepsilon}\Big[g\big(m_{P_\varepsilon}(y)\big)\Big] = (1 - \varepsilon)\mathbb{E}_{y \sim P}\Big[g\big(m_{P_\varepsilon}(y)\big)\Big] + \varepsilon\, g\big(m_{P_\varepsilon}(\tau)\big).$$

Decomposing $\mathcal{F}(P)$ into an $\epsilon$-part and another $(1 - \epsilon)$-part and rearranging yields the exact identity

$$\mathcal{F}(P_\varepsilon) - \mathcal{F}(P) = \underbrace{\varepsilon\Big(g\big(m_{P_\varepsilon}(\tau)\big) - \mathcal{F}(P)\Big)}_{\text{main term: contribution of injected } \tau} + \underbrace{(1 - \varepsilon)\,\mathbb{E}_{y \sim P}\big[g\big(m_{P_\varepsilon}(y)\big) - g\big(m_P(y)\big)\big]}_{\text{perturbation on existing samples}}. \tag{11}$$

Using $m_{P_\varepsilon}(y) = m_P(y) + \varepsilon d_\tau(y)$ from (10),

$$\mathcal{F}(P_\varepsilon) - \mathcal{F}(P) = \varepsilon\Big(g\big(m_{P_\varepsilon}(\tau)\big) - \mathcal{F}(P)\Big) + (1 - \varepsilon)\,\mathbb{E}_{y \sim P}\big[g\big(m_P(y) + \varepsilon d_\tau(y)\big) - g\big(m_P(y)\big)\big]. \tag{12}$$

## C.2. Infinitesimal Analysis: The Influence Function

We now analyze how $\mathcal{F}$ responds to infinitesimal changes. This characterizes the "gradient" of diversity in the space of distributions, indicating which trajectories should be upweighted to maximally increase diversity.

Consider the mixture perturbation $P_\varepsilon := (1 - \varepsilon)P + \varepsilon\,\delta_\tau$ in the limit as $\varepsilon \to 0^+$.

**Theorem C.2** (Diversity Influence Function). *The Gâteaux derivative of $\mathcal{F}$ at $P$ in the direction of a trajectory $\tau$ is given by:*

$$\mathcal{I}(\tau; P) := \frac{d}{d\varepsilon}\mathcal{F}(P_\varepsilon)\bigg|_{\varepsilon=0} = \underbrace{g(m_P(\tau))}_{\text{Intrinsic Novelty}} + \underbrace{\mathbb{E}_{y \sim P}\left[g'(m_P(y))k(y, \tau)\right]}_{\text{Interaction Cost}} - \Psi(P), \tag{13}$$

*where $\Psi(P)$ is a scalar baseline independent of $\tau$.*

*Proof.* Recall the exact decomposition of the finite difference derived in (12):

$$\mathcal{F}(P_\varepsilon) - \mathcal{F}(P) = \varepsilon\Big(g\big(m_{P_\varepsilon}(\tau)\big) - \mathcal{F}(P)\Big) + (1 - \varepsilon)\,\mathbb{E}_{y \sim P}\big[g\big(m_P(y) + \varepsilon d_\tau(y)\big) - g\big(m_P(y)\big)\big].$$

Dividing both sides by $\varepsilon$ and taking the limit $\varepsilon \to 0^+$, we analyze the two terms on the right-hand side separately.

**The Injection Term.** By the continuity of the kernel, $\lim_{\varepsilon \to 0} m_{P_\varepsilon}(\tau) = m_P(\tau)$. Thus, the first term becomes:

$$\lim_{\varepsilon \to 0^+} \frac{\varepsilon\Big(g\big(m_{P_\varepsilon}(\tau)\big) - \mathcal{F}(P)\Big)}{\varepsilon} = g\big(m_P(\tau)\big) - \mathcal{F}(P). \tag{14}$$

**The Perturbation Term.** For the second term, we apply the limit to the expectation. Since $g$ is continuously differentiable and the domain is bounded, the Dominated Convergence Theorem allows us to interchange the limit and the integral:

$$\lim_{\varepsilon \to 0^+}(1 - \varepsilon)\mathbb{E}_{y \sim P}\left[\frac{g\big(m_P(y) + \varepsilon d_\tau(y)\big) - g\big(m_P(y)\big)}{\varepsilon}\right] = 1 \cdot \mathbb{E}_{y \sim P}\left[\frac{d}{d\varepsilon}\Big[g\big(m_P(y) + \varepsilon d_\tau(y)\big)\Big]_{\varepsilon=0}\right]$$

$$= \mathbb{E}_{y \sim P}\Big[g'\big(m_P(y)\big) \cdot d_\tau(y)\Big]. \tag{15}$$

Substituting the definition of the similarity offset $d_\tau(y) = k(y, \tau) - m_P(y)$ into (15), we expand the interaction:

$$\mathbb{E}_{y \sim P}\Big[g'\big(m_P(y)\big)\big(k(y, \tau) - m_P(y)\big)\Big] = \mathbb{E}_{y \sim P}\Big[g'\big(m_P(y)\big)k(y, \tau)\Big] - \mathbb{E}_{y \sim P}\Big[g'\big(m_P(y)\big)m_P(y)\Big].$$

Summing the results from the injection and perturbation terms:

$$\mathcal{I}(\tau; P) = \Big(g(m_P(\tau)) - \mathcal{F}(P)\Big) + \Big(\mathbb{E}_{y \sim P}[g'(m_P(y))k(y,\tau)] - \mathbb{E}_{y \sim P}[g'(m_P(y))m_P(y)]\Big)$$

$$= g(m_P(\tau)) + \mathbb{E}_{y \sim P}[g'(m_P(y))k(y,\tau)] - \underbrace{\Big(\mathcal{F}(P) + \mathbb{E}_{y \sim P}[g'(m_P(y))m_P(y)]\Big)}_{\Psi(P)}.$$

The term $\Psi(P)$ depends solely on the reference distribution $P$ and is constant with respect to $\tau$. $\qquad\square$

To determine the correlation between $\mathcal{I}(\tau; P)$ and $m_P(\tau)$, we show that provided $g$ is strictly decreasing, the interaction mechanism inherently cooperates with the intrinsic novelty to penalize redundancy.

Since $\Psi(P)$ is independent of $\tau$, for ranking or gradient-based reweighting it is often sufficient to use the uncentered score

$$\tilde{\mathcal{I}}(\tau; P) := g(m_P(\tau)) + \mathbb{E}_{y \sim P}[g'(m_P(y))\, k(y,\tau)].$$

The interaction term depends on $m_P(y)$ in the neighborhood of $\tau$ weighted by $k(y,\tau)$. To obtain a local monotonicity statement without a fragile *relative* homogeneity assumption, we use an *absolute* smoothness control on an effective neighborhood and explicitly account for the kernel tail.

**Assumption C.3** (Absolute local smoothness at kernel threshold $\eta$). Fix a threshold $\eta \in (0,1)$. For $\tau \in \Omega$, define the effective neighborhood

$$B_{\tau,\eta} := \{y \in \Omega : \; k(y,\tau) \geq \eta\},$$

and assume there exists $\delta = \delta(\tau,\eta) \geq 0$ such that

$$\sup_{y \in B_{\tau,\eta}} |m_P(y) - m_P(\tau)| \leq \delta(\tau,\eta).$$

**Lemma C.4** (Proxy reduction with an explicit remainder bound). *Under Assumption C.1 and C.3, then for any $\eta \in (0,1)$ and any $\tau \in \Omega$,*

$$\tilde{\mathcal{I}}(\tau; P) = \phi(m_\tau) + r_\tau, \qquad \phi(m) := g(m) + mg'(m), \quad \text{where} \quad m_\tau := m_P(\tau),$$

*and the remainder satisfies*

$$|r_\tau| \leq L\left(\delta(\tau,\eta)\, m_\tau \; + \; \eta\right). \tag{16}$$

*Proof.* Write

$$\tilde{\mathcal{I}}(\tau; P) = g(m_\tau) + \mathbb{E}[g'(m_P(y))k(y,\tau)]$$
$$= g(m_\tau) + g'(m_\tau)\mathbb{E}[k(y,\tau)] + \mathbb{E}[(g'(m_P(y)) - g'(m_\tau))k(y,\tau)].$$

Since $\mathbb{E}[k(y,\tau)] = m_\tau$, the first two terms equal $\phi(m_\tau)$. Thus the remainder is

$$r_\tau = \mathbb{E}[(g'(m_P(y)) - g'(m_\tau))k(y,\tau)].$$

Split the expectation over $B_{\tau,\eta}$ and its complement. On $B_{\tau,\eta}$, Assumption C.3 and lipschitzness of $g'$ give $|g'(m_P(y)) - g'(m_\tau)| \leq L|m_P(y) - m_\tau| \leq L\delta$, hence

$$\left|\mathbb{E}\big[(g'(m_P(y)) - g'(m_\tau))k(y,\tau)\mathbf{1}_{B_{\tau,\eta}}\big]\right| \leq L\delta\, \mathbb{E}[k(y,\tau)] = L\delta\, m_\tau.$$

On $B_{\tau,\eta}^c$, we have $k(y,\tau) < \eta$ and $|g'(m_P(y)) - g'(m_\tau)| \leq L$, hence

$$\left|\mathbb{E}\Big[(g'(m_P(y)) - g'(m_\tau))k(y,\tau)\mathbf{1}_{B_{\tau,\eta}^c}\Big]\right| \leq L\eta.$$

Combining the two bounds proves (16). $\qquad\square$

**Theorem C.5** (Local monotonicity via a curvature condition). *Assume in addition that $g$ is twice continuously differentiable on $(0,1]$ and satisfies*

$$mg''(m) < -2g'(m), \qquad \forall m \in (0,1]. \tag{17}$$

*Moreover, fix any $a \in (0,1]$ and define*

$$c_a := \min_{m \in [a,1]} \big( -\phi'(m) \big) > 0.$$

*For any $\tau_1, \tau_2$ with $m_P(\tau_1), m_P(\tau_2) \in [a,1]$, we have*

$$\tilde{\mathcal{I}}(\tau_1; P) - \tilde{\mathcal{I}}(\tau_2; P) \le -c_a\big(m_P(\tau_1) - m_P(\tau_2)\big) + \big(|r_{\tau_1}| + |r_{\tau_2}|\big), \tag{18}$$

*where $r_\tau$ is the remainder in Lemma C.4. In particular, whenever*

$$m_P(\tau_1) - m_P(\tau_2) > \frac{|r_{\tau_1}| + |r_{\tau_2}|}{c_a},$$

*the ranking aligns with rarity:*

$$m_P(\tau_1) > m_P(\tau_2) \quad \Longrightarrow \quad \tilde{\mathcal{I}}(\tau_1; P) < \tilde{\mathcal{I}}(\tau_2; P).$$

*Thus, in regimes where the local smoothness error $\delta(\tau, \eta)$ and tail threshold $\eta$ are small, the interaction term cooperates with the intrinsic novelty term to penalize redundancy.*

*Proof.* The strict decrease of $\phi$ follows from $\phi'(m) = 2g'(m) + mg''(m)$ and (17). On the compact interval $[a,1]$, $\phi'$ is continuous and negative, hence $c_a = \min_{m \in [a,1]}(-\phi'(m)) > 0$. By the mean value theorem, for $m_1, m_2 \in [a,1]$,

$$\phi(m_1) - \phi(m_2) \le -c_a(m_1 - m_2).$$

Apply this to $m_i = m_P(\tau_i)$ and use the decomposition (C.4):

$$\tilde{\mathcal{I}}(\tau_1; P) - \tilde{\mathcal{I}}(\tau_2; P) = \phi(m_1) - \phi(m_2) + (r_{\tau_1} - r_{\tau_2}) \le -c_a(m_1 - m_2) + |r_{\tau_1}| + |r_{\tau_2}|,$$

which is (18). $\qquad \square$

**Corollary C.6** (Soft-log stability and bounded gradients). *For $g(m) = -\log(1+m)$, the curvature condition (17) holds for all $m \ge 0$. Moreover, $|g'(m)| \le 1$ on $[0,1]$, hence the influence score avoids gradient blow-up in low-mass regions.*

*Proof.* For $g(m) = -\log(1+m)$, we have $g'(m) = -(1+m)^{-1}$ and $g''(m) = (1+m)^{-2}$, thus

$$\phi'(m) = 2g'(m) + mg''(m) = \frac{-2(1+m) + m}{(1+m)^2} = \frac{-(m+2)}{(1+m)^2} < 0,$$

which implies (17). Also $|g'(m)| = (1+m)^{-1} \le 1$ on $[0,1]$. $\qquad \square$

## C.3. Finite Perturbation Analysis and Error Bounds

While the influence function characterizes the optimal direction for infinitesimal updates ($\varepsilon \to 0^+$), practical generation algorithms operate with discrete perturbation ($\varepsilon > 0$). In this subsection, we establish rigorous bounds for the diversity gain under finite perturbations. We demonstrate that the trajectory redundancy $m_P(\tau)$ governs both the magnitude of the expected gain and the tightness of the approximation error.

**Theorem C.7** (Finite Perturbation Bounds). *Denote $M := \max_{x \in [0,1]} |g'(x)|$. For any trajectory $\tau \in \Omega$ and perturbation scale $\varepsilon \in [0,1]$, the finite change in diversity $\Delta\mathcal{F} = \mathcal{F}(P_\varepsilon) - \mathcal{F}(P)$ is bounded by:*

$$\mathcal{M}(\tau, \varepsilon) - \mathcal{E}(\tau, \varepsilon) \;\le\; \Delta\mathcal{F} \;\le\; \mathcal{M}(\tau, \varepsilon) + \mathcal{E}(\tau, \varepsilon),$$

*where the **main signal** $\mathcal{M}$ and the **error radius** $\mathcal{E}$ are defined as:*

$$\mathcal{M}(\tau, \varepsilon) := \varepsilon \big( g\big((1-\varepsilon)m_P(\tau) + \varepsilon\big) - \mathcal{F}(P) \big),$$
$$\mathcal{E}(\tau, \varepsilon) := M \cdot \varepsilon \cdot \big( m_P(\tau) + \kappa(P) \big).$$

*Here, $\kappa(P) := \mathbb{E}_{y, y' \sim P}[k(y, y')]$ represents the global average similarity of the distribution.*

*Proof.* Recall the exact decomposition of the finite difference derived in Eq. (11):

$$\Delta\mathcal{F} = \mathcal{M}(\tau, \varepsilon) + B(\varepsilon), \tag{19}$$

where the residual term is $B(\varepsilon) = (1 - \varepsilon)\mathbb{E}_{y\sim P}[g(m_{P_\varepsilon}(y)) - g(m_P(y))]$. To prove the theorem, it suffices to bound the absolute magnitude $|B(\varepsilon)|$.

By Assumption C.1, substitute the linear update rule $m_{P_\varepsilon}(y) = m_P(y) + \varepsilon d_\tau(y)$, where $d_\tau(y) = k(y, \tau) - m_P(y)$, we get:

$$|g(m_{P_\varepsilon}(y)) - g(m_P(y))| \le M \cdot \varepsilon \cdot |d_\tau(y)|.$$

Since the kernel is non-negative ($k \ge 0$), applying the triangle inequality directly yields:

$$|d_\tau(y)| = |k(y, \tau) - m_P(y)| \le k(y, \tau) + m_P(y).$$

Taking the expectation over $y \sim P$:

$$\mathbb{E}_{y\sim P}|d_\tau(y)| \le \mathbb{E}_{y\sim P}[k(y, \tau)] + \mathbb{E}_{y\sim P}[m_P(y)] = m_P(\tau) + \kappa(P). \tag{20}$$

Hence, it holds that

$$|B(\varepsilon)| \le (1 - \varepsilon) \cdot \mathbb{E}_{y\sim P}[M \cdot \varepsilon \cdot |d_\tau(y)|] \le M \cdot \varepsilon \cdot (m_P(\tau) + \kappa(P)) = \mathcal{E}(\tau, \varepsilon).$$

The two-sided bound follows immediately from $-|B(\varepsilon)| \le B(\varepsilon) \le |B(\varepsilon)|$. $\qquad\square$

Theorem C.7 uncovers a structural duality where the redundancy $m_P(\tau)$ governs both the potential upside and the theoretical risk of an update. Specifically, minimizing $m_P(\tau)$ simultaneously enhances the novelty signal $\mathcal{M}$ and strictly compresses the error radius $\mathcal{E}$. This implies that rarer trajectories offer a superior trade-off. They provide higher projected gains with greater certainty. To rigorously operationalize this observation, we focus on the certified lower bound, defined as $J(\tau, \varepsilon) := \mathcal{M}(\tau, \varepsilon) - \mathcal{E}(\tau, \varepsilon)$, which serves as the worst-case guarantee for improvement.

**Theorem C.8** (Monotonicity of certified gain). *If $\epsilon > 0$, then the certified lower bound $J(\tau, \varepsilon)$ is a strictly decreasing function of the local mass $m_P(\tau)$.*

*Proof.* Let $m := m_P(\tau) \in [0, 1]$. We analyze the sensitivity of the lower bound function $J$ with respect to $m$:

$$J(m) = \varepsilon\Big[g\big((1 - \varepsilon)m + \varepsilon\big) - \mathcal{F}(P) - Mm - M\kappa(P)\Big].$$

Differentiating $J(m)$ yields:

$$\frac{\partial J}{\partial m} = \varepsilon\left[(1 - \varepsilon)g'\big((1 - \varepsilon)m + \varepsilon\big) - M\right].$$

The strict negativity of this derivative follows immediately from the system's properties. $\qquad\square$

This confirms that increasing redundancy incurs a "double penalty," simultaneously attenuating the diversity signal and amplifying the perturbation noise.

## D. Discrete Analysis for Leave-One-Out Contribution $s_i$

**Definitions.** Let $\Omega = \{o_1, \dots, o_G\}$ with $G \ge 3$. Assume the similarity kernel is symmetric: $k(o_p, o_q) = k(o_q, o_p)$, and bounded on its domain. For any finite set $S$ with $|S| = n \ge 2$, define the leave-one-out local mass

$$m_u(S) = \frac{1}{|S| - 1} \sum_{v\in S,\, v\ne u} k(o_u, o_v),$$

and the diversity score

$$D(S) = \frac{1}{|S|} \sum_{u\in S} g\big(m_u(S)\big),$$

where $g$ is continuously differentiable and strictly decreasing on the relevant range: $g'(x) < 0$.

For the full set $\Omega$ we write $m_i := m_i(\Omega)$. For a leave-one-out set $A_{-i} := \Omega \setminus \{o_i\}$ we write

$$m_j^{(-i)} := m_j(A_{-i}), \qquad D_{-i} := D(A_{-i}).$$

Your algorithm uses the leave-one-out contribution

$$s_i := D(\Omega) - D_{-i}.$$

**Lemma D.1** (Exact density update law under removal). *For any $j \neq i$,*

$$m_j^{(-i)} = \frac{1}{G-2} \sum_{p \in \Omega, \, p \neq j, i} k(o_j, o_p) = \frac{(G-1)m_j - k(o_j, o_i)}{G-2} = m_j + \frac{m_j - k(o_j, o_i)}{G-2}.$$

*Equivalently,*

$$m_j^{(-i)} - m_j = \frac{m_j - k(o_j, o_i)}{G-2}.$$

*Proof.* Let $R_{ji} := \sum_{p \in \Omega, \, p \neq j, i} k(o_j, o_p)$. Then $m_j = \frac{1}{G-1}\big(k(o_j, o_i) + R_{ji}\big)$ and $m_j^{(-i)} = \frac{1}{G-2}R_{ji}$. Eliminate $R_{ji} = (G-1)m_j - k(o_j, o_i)$ and substitute. $\square$

**Theorem D.2** (Exact decomposition of $s_i$). *Let $G = |\Omega| \geq 3$. Then*

$$s_i = \underbrace{\frac{1}{G}g(m_i)}_{\mathcal{T}_{self}} + \underbrace{\left(\frac{1}{G} - \frac{1}{G-1}\right) \sum_{j \neq i} g(m_j)}_{\mathcal{T}_{norm}} + \underbrace{\frac{1}{G-1} \sum_{j \neq i} \left[g(m_j) - g(m_j^{(-i)})\right]}_{\mathcal{T}_{interaction}}.$$

*Moreover, by Lemma D.1, each interaction summand depends on $k(o_j, o_i)$ only through $m_j^{(-i)} = m_j + \frac{m_j - k(o_j, o_i)}{G-2}$.*

*Proof.* Expand $D(\Omega) = \frac{1}{G}\big(g(m_i) + \sum_{j \neq i} g(m_j)\big)$ and $D_{-i} = \frac{1}{G-1} \sum_{j \neq i} g(m_j^{(-i)})$ and regroup terms. $\square$

**Theorem D.3** (Strict monotonicity w.r.t. pairwise similarity). *Fix $i \neq t$ and denote $k_{it} := k(o_i, o_t)$. Then*

$$\frac{\partial s_i}{\partial k_{it}} = \frac{1}{G(G-1)}\Big(g'(m_i) + g'(m_t)\Big) < 0.$$

*Proof.* $D_{-i}$ does not involve $o_i$, hence is independent of $k_{it}$. Therefore $\frac{\partial s_i}{\partial k_{it}} = \frac{\partial D(\Omega)}{\partial k_{it}}$. In $D(\Omega)$, only $m_i$ and $m_t$ depend on $k_{it}$, and $\frac{\partial m_i}{\partial k_{it}} = \frac{1}{G-1}$, $\frac{\partial m_t}{\partial k_{it}} = \frac{1}{G-1}$. Thus

$$\frac{\partial D(\Omega)}{\partial k_{it}} = \frac{1}{G}\left(g'(m_i)\frac{1}{G-1} + g'(m_t)\frac{1}{G-1}\right) = \frac{1}{G(G-1)}(g'(m_i) + g'(m_t)) < 0,$$

since $g' < 0$. $\square$

**Theorem D.4** (Rare–redundant ordering under pointwise dominance). *Let $r \neq c$ be two indices in $\Omega$. Assume $o_r$ is pointwise no-more-similar than $o_c$ to all other points:*

$$k(o_r, o_j) \leq k(o_c, o_j) \quad \forall j \in \Omega \setminus \{r, c\},$$

*with strict inequality for at least one such $j$. Then*

$$s_r > s_c.$$

*Proof.* Since $s_i = D(\Omega) - D_{-i}$ and $D(\Omega)$ is common,

$$s_r - s_c = D_{-c} - D_{-r}.$$

Expand

$$D_{-c} = \frac{1}{G-1}\left(g(m_r^{(-c)}) + \sum_{j \neq r,c} g(m_j^{(-c)})\right), \quad D_{-r} = \frac{1}{G-1}\left(g(m_c^{(-r)}) + \sum_{j \neq r,c} g(m_j^{(-r)})\right),$$

hence

$$s_r - s_c = \frac{1}{G-1}\left(g(m_r^{(-c)}) - g(m_c^{(-r)}) + \sum_{j \neq r,c} \left[g(m_j^{(-c)}) - g(m_j^{(-r)})\right]\right).$$

Now note:

$$m_r^{(-c)} = \frac{1}{G-2}\sum_{j \neq r,c} k(o_r, o_j), \quad m_c^{(-r)} = \frac{1}{G-2}\sum_{j \neq r,c} k(o_c, o_j),$$

hence $m_r^{(-c)} \leq m_c^{(-r)}$ by the assumption, with strict inequality if any term is strict. For $j \neq r, c$, apply Lemma D.1 to both removals and subtract:

$$m_j^{(-c)} - m_j^{(-r)} = \frac{(G-1)m_j - k(o_j, o_c)}{G-2} - \frac{(G-1)m_j - k(o_j, o_r)}{G-2} = \frac{k(o_j, o_r) - k(o_j, o_c)}{G-2} \leq 0,$$

again strict for at least one $j$ if the dominance is strict somewhere (using symmetry of $k$). Since $g$ is strictly decreasing, every bracketed difference in the expansion of $s_r - s_c$ is nonnegative, and at least one is strictly positive, so $s_r - s_c > 0$. $\qquad\square$

**Corollary D.5** (Weighted characterization of the gap). *There exist $\xi_0$ between $m_r^{(-c)}$ and $m_c^{(-r)}$, and $\{\xi_j\}_{j \neq r,c}$ between $m_j^{(-c)}$ and $m_j^{(-r)}$, such that*

$$s_r - s_c = \frac{1}{(G-1)(G-2)}\sum_{j \neq r,c}\left(w_0 + w_j\right)\Big(k(o_c, o_j) - k(o_r, o_j)\Big), \quad w_0 := -g'(\xi_0) > 0, \quad w_j := -g'(\xi_j) > 0.$$

*Therefore $s_r > s_c$ holds whenever the weighted similarity advantage of $o_c$ over $o_r$ is positive, even if pointwise dominance fails.*

*Proof.* Apply the mean value theorem to $g(m_r^{(-c)}) - g(m_c^{(-r)})$ and each $g(m_j^{(-c)}) - g(m_j^{(-r)})$ in the expansion of Theorem D.4, then use

$$m_r^{(-c)} - m_c^{(-r)} = \frac{1}{G-2}\sum_{j \neq r,c}(k(o_r, o_j) - k(o_c, o_j)), \quad m_j^{(-c)} - m_j^{(-r)} = \frac{k(o_j, o_r) - k(o_j, o_c)}{G-2},$$

and simplify. $\qquad\square$

**Corollary D.6** (Dilution vs. congestion under removal). *For any $j \neq i$, removing $o_i$ makes $o_j$ denser (mass increases) iff $o_i$ acts as a diluter for $o_j$:*

$$m_j^{(-i)} > m_j \iff k(o_j, o_i) < m_j.$$

*Consequently, the neighbor score decreases under removal iff $k(o_j, o_i) < m_j$:*

$$g(m_j^{(-i)}) < g(m_j) \iff k(o_j, o_i) < m_j.$$

*Proof.* Immediate from Lemma D.1 and monotonicity of $g$. $\qquad\square$

# E. Experiments Setting

## E.1. Training Settings

We train Qwen-1.5B and Qwen-7B models on GSM8K following the experimental settings in (Yao et al., 2025) to ensure a fair comparison. The hyperparameters for SetPO+GRPO, SetPO+GSPO, and SetPO+DAPO are summarized in Tables 4, 5, and 6, respectively. For the extended experiments on Qwen-32B, the detailed settings are provided in Table 7. For the Countdown experiments, the corresponding hyperparameters are listed in Table 8. In all settings, the kernel function is defined as $k(o_i, o_j) = \text{clamp}(\text{sim}(e_i, e_j))$, where $o_i$ and $o_j$ denote the output answers and $e_i, e_j$ are their corresponding semantic embeddings. Here, $\text{sim}$ represents cosine similarity, and the $\text{clamp}$ operation is applied to ensure the kernel value remains within the range $[0, 1]$.

*Table 4.* Hyperparameters and Experimental Settings for SetPO+GRPO and GRPO

| Parameter | Value | Parameter | Value |
|---|---|---|---|
| Base Model | Qwen2.5-Math-7B/1.5B | Algorithm | GRPO |
| Dataset | GSM8K | Embedding Model | Qwen3-Embedding-0.6B |
| *Training Configuration* | | | |
| Global Batch Size | 64 | Learning Rate | $3 \times 10^{-6}$ |
| KL Coefficient | $1 \times 10^{-4}$ | Total Epochs | 2 |
| Rollout Samples ($N$) | 6 | Training Rollout Temp. | 0.9 |
| Clip Ratio | 0.2 | Max Response Len. | 756 |
| Marginal Weight | 0.05 | Shape Function | $-\log(1 + x)$ |
| *Inference & Evaluation Configuration* | | | |
| Inference Temperature | 0.5 | Top-$p$ | 0.9 |

*Table 5.* Hyperparameters and Experimental Settings for SetPO+GSPO and GSPO

| Parameter | Value | Parameter | Value |
|---|---|---|---|
| Base Model | Qwen2.5-Math-7B/1.5B | Algorithm | GSPO |
| Dataset | GSM8K | Embedding Model | Qwen3-Embedding-0.6B |
| *Training Configuration* | | | |
| Global Batch Size | 64 | Learning Rate | $1 \times 10^{-6}$ |
| KL Coefficient | $1 \times 10^{-4}$ | Total Epochs | 2 |
| Clip Ratio Low ($\epsilon_{low}$) | 0.0003 | Clip Ratio High ($\epsilon_{high}$) | 0.0004 |
| Temperature | 0.9 | Max Response Len. | 756 |
| Rollout Number | 6 | Mini Batch Size | 64 |
| Marginal Weight | 0.1 | Shape Function | $-\log(1 + x)$ |
| *Inference & Evaluation Configuration* | | | |
| Inference Temperature | 0.5 | Top-$p$ | 0.9 |

## E.2. Evaluation Settings

Following the evaluation protocol (Chen, 2021), we employ the unbiased estimator for the Pass@k metric. Directly generating only $k$ samples to measure accuracy often yields high variance. Therefore, we generate a larger pool of $n$ samples and analytically calculate the probability of finding at least one correct solution within a random subset of size $k$. Let $n$ denote the total sample count and $c$ the count of correct solutions. The unbiased estimator is defined as:

$$\text{Pass@}k := 1 - \frac{\binom{n-c}{k}}{\binom{n}{k}} \tag{21}$$

*Table 6.* Hyperparameters and Experimental Settings for SetPO+DAPO and DAPO

| Parameter | Value | Parameter | Value |
|---|---|---|---|
| Base Model | Qwen2.5-Math-7B/1.5B | Algorithm | DAPO |
| Dataset | GSM8K | Embedding Model | Qwen3-Embedding-0.6B |
| *Training Configuration* | | | |
| Global Batch Size | 64 | Learning Rate | $1 \times 10^{-6}$ |
| KL Coefficient | $1 \times 10^{-4}$ | Total Epochs | 2 |
| Clip Ratio Low ($\epsilon_{low}$) | 0.2 | Clip Ratio High ($\epsilon_{high}$) | 0.28 |
| Temperature | 0.9 | Max Response Len. | 756 |
| Rollout Number | 6 | Mini Batch Size | 64 |
| Marginal Weight | 0.2 | Shape Function | $-\log(1+x)$ |
| *Inference & Evaluation Configuration* | | | |
| Inference Temperature | 0.5 | Top-$p$ | 0.9 |

*Table 7.* Hyperparameters and Experimental Settings for SetPO+GRPO and GRPO of Qwen2.5-32B-Base

| Parameter | Value | Parameter | Value |
|---|---|---|---|
| Base Model | Qwen2.5-32B-Base | Algorithm | GRPO |
| Dataset | Math-Hard (Zeng et al., 2025) | Embedding Model | Qwen3-Embedding-0.6B |
| *Training Configuration* | | | |
| Global Batch Size | 1024 | Learning Rate | $1 \times 10^{-6}$ |
| KL Coefficient | $1 \times 10^{-3}$ | Total Epochs | 15 |
| Clip Ratio | 0.2 | Mini Batch Size | 256 |
| Temperature | 1 | Max Response Len. | 4096 |
| Marginal Weight | 0.1 | Shape Function | $-\log(1+x)$ |
| Rollout Number | 6 | Mini Batch Size | 64 |
| *Inference & Evaluation Configuration* | | | |
| Inference Temperature | 0.5 | Top-$p$ | 0.9 |

*Table 8.* Detailed hyperparameter settings for the Qwen2.5-3B GRPO experiments for Countdown.

| Parameter | Value | Parameter | Value |
|---|---|---|---|
| Base Model | Qwen2.5-3B | Algorithm | GRPO |
| Dataset | Countdown | Embedding Model | Qwen3-Embedding-0.6B |
| *Training Configuration* | | | |
| Global Batch Size | 128 | Learning Rate | $1 \times 10^{-6}$ |
| KL Coefficient ($\beta$) | 0.001 | Training Epochs | 1 |
| Rollout Group Size ($G$) | 7/ 3 | Total Training Samples | 86400 |
| Max Prompt Length | 512 | Max Response Length | 2048 |
| Marginal Weight | 0.5 | Shape Function | $-\log(1+x)$ |
| *Inference & Evaluation Configuration* | | | |
| Eval Response Length | 2048 | Eval Samples ($N$) | 64 |
| Temperature | 1.0/ 0.5 | Top-$p$ | 0.9 |

In our experiments, we fix the total generation budget at $n = 64$ and evaluate Pass@k for $k \in \{1, 2, 4, 8, 16, 32, 64\}$. This formula naturally simplifies to empirical accuracy when $n = k$. Additionally, if the number of incorrect samples is fewer than $k$ (i.e., $n - c < k$), the metric is defined as 1, indicating that any chosen subset of size $k$ would inherently contain a correct solution.

## F. More Results on Mathematical Reasoning

### F.1. Complete Comparison on Mathematical Benchmark

The complete comparison from Pass@1 to Pass@k is shown in Figures 6 and 7. We evaluate different sampling budgets for the three baselines and their SetPO-augmented variants. All experiments are conducted with five random seeds to reduce randomness, and the shaded regions indicate variance. We observe that SetPO consistently reduces the variance of the baselines and yields stable performance improvements. Moreover, the gains do not diminish as the sampling budget increases, suggesting that SetPO effectively improves outcome diversity by covering more reward modes. Finally, Figure 10 further confirms that our method remains effective when scaling to larger models.

### F.2. Reaching Higher Performance when $K$ is sufficiently high

As noted by (Yue et al., 2025), improvements in Pass@1 do not always reflect genuine capability gains. In some cases, comparable performance can be obtained simply by increasing the sampling budget of the base model. Furthermore, they show that models trained with current policy gradient methods can underperform the base model in terms of Pass@k, suggesting that the apparent gains from RLVR may partially arise from a reduction in outcome diversity rather than improved underlying competence. Motivated by this observation, we evaluate our model on AIME24 from pass@1 through pass@256. As shown in Figure 8, our method consistently and substantially outperforms the base model across the entire range of $K$, including at $K = 256$, indicating that the gains are not merely an artifact of more sampling but reflect robust improvements over the underlying base model.

### F.3. Timing Cost on Adding SetPO to Original Algorithms

Although SetPO leverages an additional embedding model (Qwen3-0.6B-Embedding) to compute the leave-one-out marginal reward, the extra computation is lightweight and readily controllable. Figure 9 reports the end-to-end wall-clock time of baseline algorithms and their SetPO-augmented counterparts. Across GRPO, GSPO, and DAPO, incorporating SetPO increases total runtime by less than 10%, indicating that the embedding-based marginal reward estimation does not materially affect training throughput. Moreover, this overhead is not fundamental and can be further reduced with more efficient implementations of the embedding pipeline (e.g., optimized parallelization and better IO/serialization), making SetPO a practical drop-in augmentation for existing RL pipelines.

## G. Case Study

### G.1. Case Study: Diversity in Mathematical Reasoning Solutions

In this subsection, we use a representative olympiad-style number theory problem to illustrate the diversity of reasoning paths produced by our SetPO+GRPO training.

**Problem Statement**. Let $b \geq 2$ be an integer. Call a positive integer $n$ *b-eautiful* if it has exactly two digits when expressed in base $b$ and these two digits sum to $\sqrt{n}$. For example, 81 is 13-*eautiful* because $81 = \underline{6}\ \underline{3}_{13}$ and $6 + 3 = \sqrt{81}$. Find the least integer $b \geq 2$ for which there are more than ten *b-eautiful* integers.

Instead of converging to a single proof, the trained model can generate multiple solution strategies for the problem and arrive at the same final answer. Below, we present two such solutions: one proceeds mainly via algebraic manipulation of the digit constraints, while the other follows a constructive viewpoint. These distinct approaches provides concrete evidence that SetPO+GRPO encourages diverse problem-solving strategies, validating the diversity of our generated solutions.

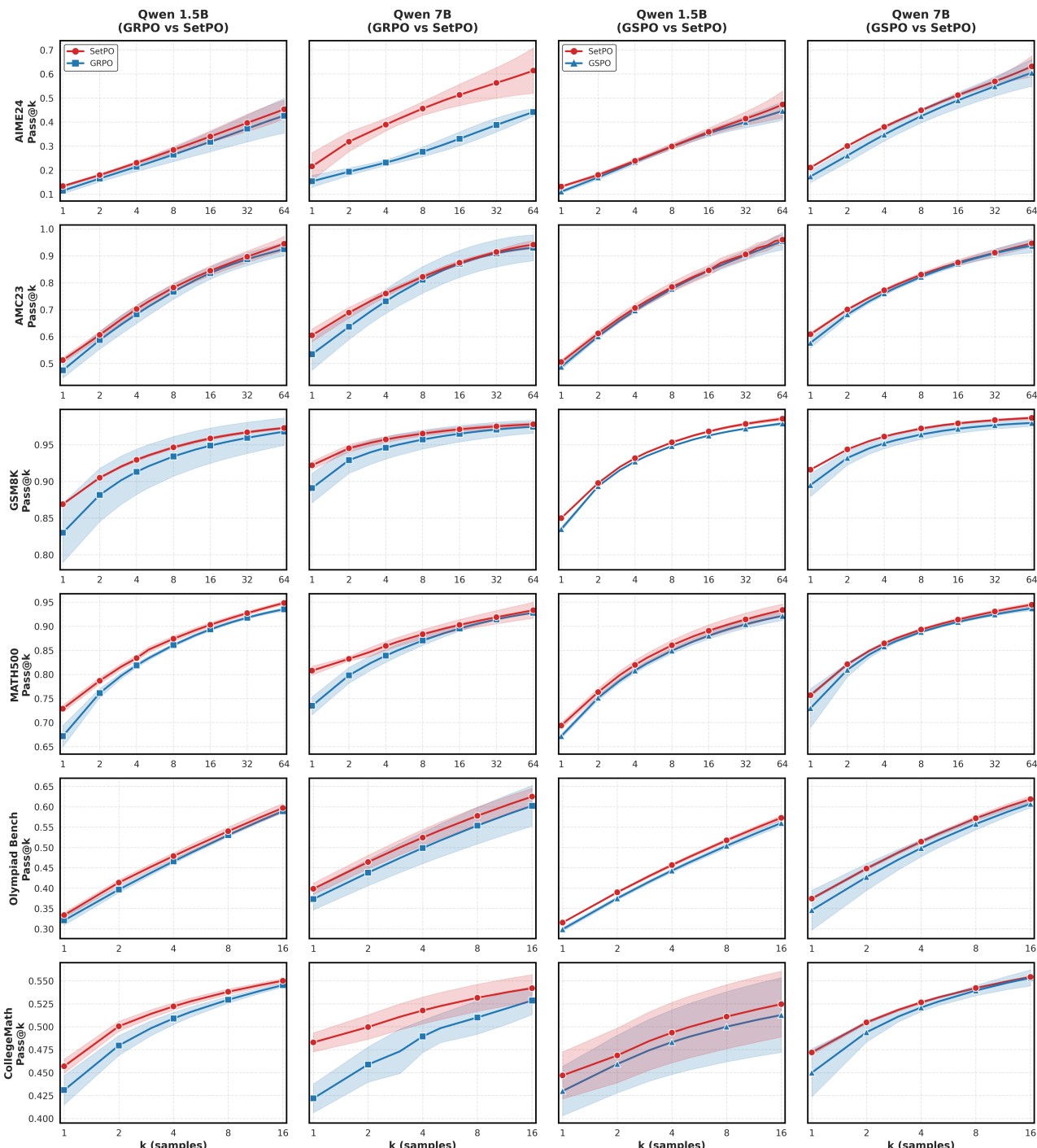

*Figure 6.* Performance of SetPO+GRPO vs. GRPO and SetPO+GSPO vs. GSPO on Qwen2.5-Math-1.5B and Qwen2.5-Math-7B across multiple benchmarks. SetPO consistently outperforms the corresponding baselines and, in most cases, reduces training variability. Shaded regions denote variance across runs, and solid lines indicate the mean.

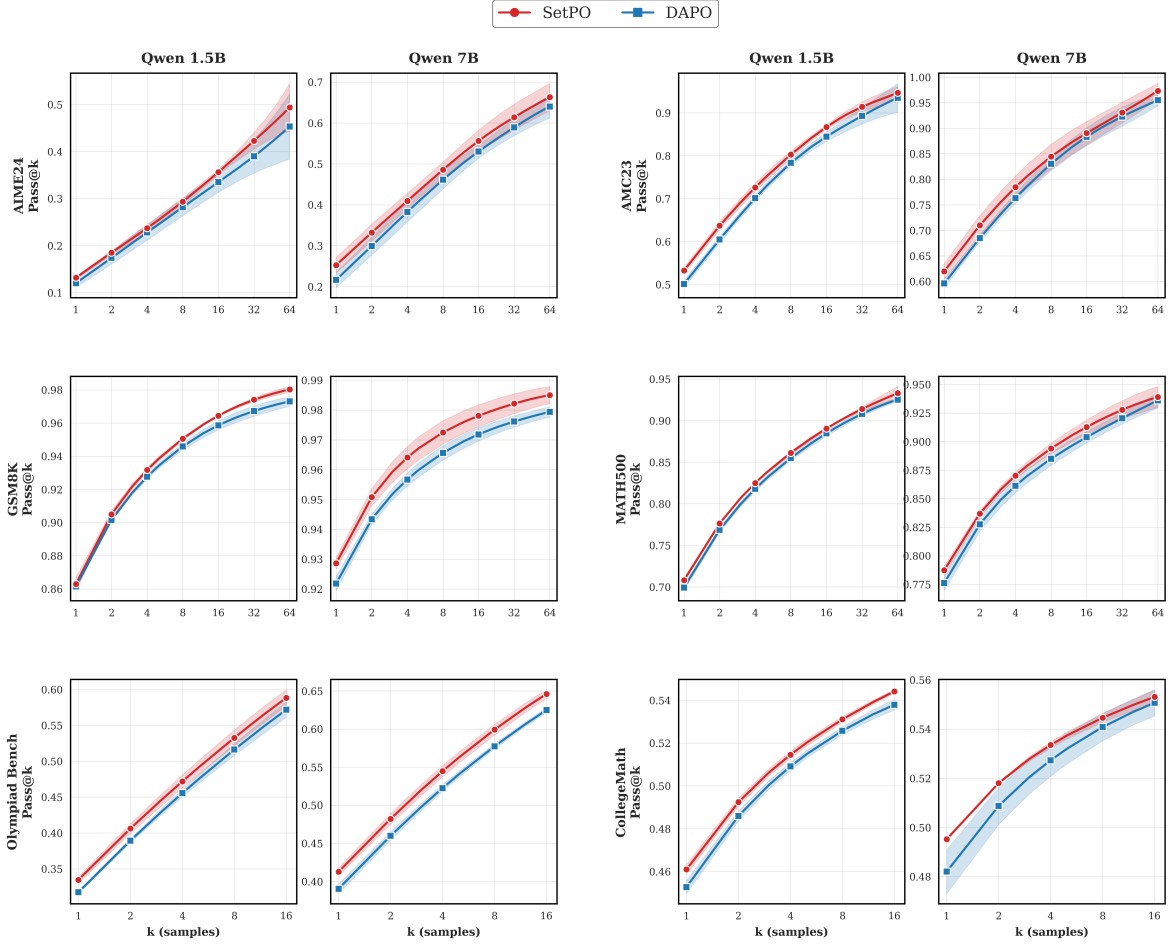

*Figure 7.* Performance of SetPO+DAPO vs. DAPO on Qwen2.5-Math-1.5B and Qwen2.5-Math-7B across multiple benchmarks. SetPO consistently outperforms the corresponding baselines. Shaded regions denote variance across runs, and solid lines indicate the mean.

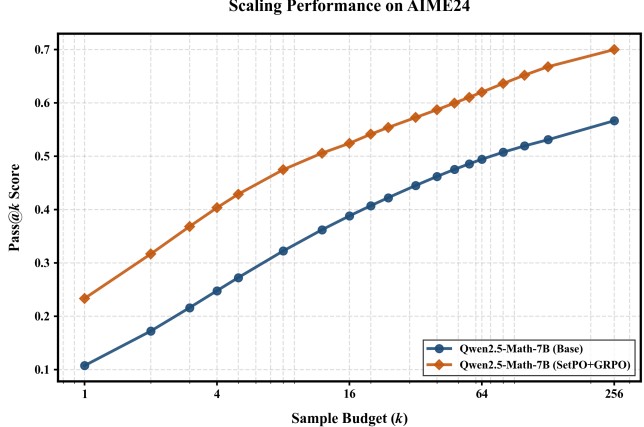

*Figure 8.* Pass@k performance on AIME24 benchmark across sampling budgets $k$, comparing the instruction-tuned base model and the RL-finetuned model (SetPO+GRPO). Notably, SetPO+GRPO maintains strong performance at large $k$, exceeding the base model even under high-pass evaluation.

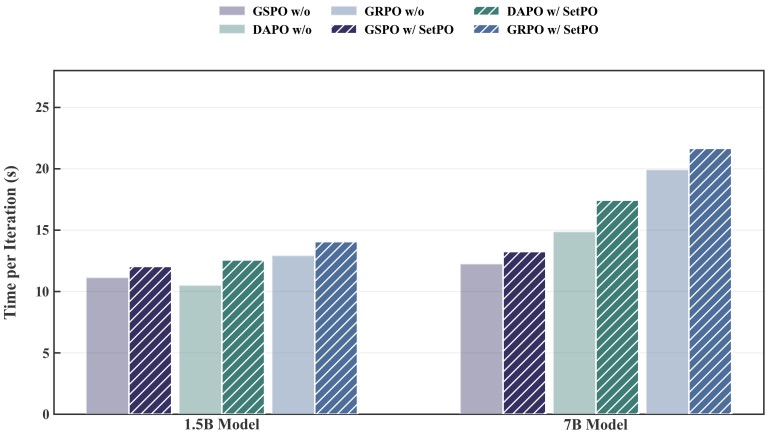

*Figure 9.* Wall-clock time comparison between baseline algorithms and their SetPO-augmented variants. For GRPO, GSPO, and DAPO, adding SetPO introduces less than 10% additional compute overhead, and this cost can be further reduced with more efficient implementations.

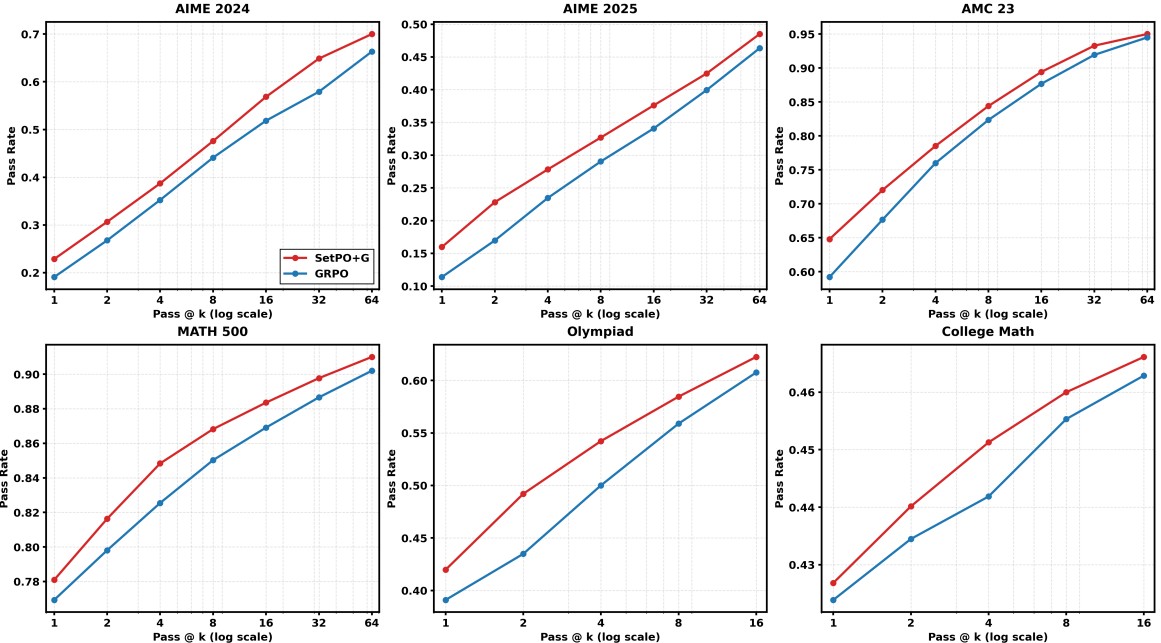

*Figure 10.* Performance of SetPO+GRPO vs GRPO on Qwen2.5-32B.

---

**Solution I**

To solve this problem, we need to find the smallest base $b \geq 2$ such that there are more than ten $b$-beautiful integers. A number $n$ is $b$-beautiful if it has exactly two digits in base $b$ and the sum of these digits is equal to $\sqrt{n}$. Let's break down the problem:

1. A two-digit number in base $b$ can be written as $n = ab + c$, where $a$ and $c$ are the digits, $1 \leq a \leq b - 1$, and $0 \leq c \leq b - 1$.

2. The condition for $n$ to be $b$-beautiful is $a + c = \sqrt{n}$. Since $n = ab + c$, we have $\sqrt{ab + c} = a + c$. Squaring both sides, we get $ab + c = (a + c)^2 = a^2 + 2ac + c^2$.

3. Rearranging the equation, we get $ab + c = a^2 + 2ac + c^2$, which simplifies to $ab = a^2 + 2ac + c^2 - c$. This can be further simplified to $ab = a^2 + c(2a + c - 1)$.

We need to find the smallest base $b$ such that there are more than ten solutions to this equation. We'll start with $b = 2$ and increment $b$ until we find a base with more than ten $b$-beautiful integers. The least integer $b \geq 2$ for which there are more than ten $b$-beautiful integers is $\boxed{211}$.

---

**Solution II**

Let's break down the problem step by step and use to find the solution.

1. **Understanding $b$-beautiful numbers**: A number $n$ is $b$-beautiful if it has exactly two digits in base $b$ and the sum of these digits is $\sqrt{n}$. If $n$ has two digits in base $b$, it can be expressed as $n = d_1 \cdot b + d_2$ where $d_1$ and $d_2$ are the digits, with $1 \leq d_1 \leq b - 1$ and $0 \leq d_2 \leq b - 1$. The condition for $n$ being $b$-beautiful is $d_1 + d_2 = \sqrt{n}$.

2. **Range of $n$**: Since $n$ has two digits in base $b$, $b \leq n < b^2$. Therefore, $b \leq n < b^2$ and $\sqrt{n}$ must be an integer. This means $n$ must be a perfect square, say $n = k^2$, where $b \leq k^2 < b^2$. Hence, $k$ must be an integer such that $\sqrt{b} \leq k < b$.

3. **Finding $b$-beautiful numbers**: For each $k$ in the range $\lceil \sqrt{b} \rceil \leq k < b$, we need to check if $k^2$ can be expressed as $d_1 \cdot b + d_2$ with $d_1 + d_2 = k$. This means $d_1 = \lfloor \frac{k^2}{b} \rfloor$ and $d_2 = k^2 \mod b$. We need to check if $d_1 + d_2 = k$.

4. **Counting $b$-beautiful numbers**: We need to find the smallest $b$ such that there are more than ten $b$-beautiful numbers. We will iterate over $b$ starting from 2 and count the number of $b$-beautiful numbers for each $b$. Let's implement this in to find the solution. The least integer $b \geq 2$ for which there are more than ten $b$-beautiful integers is $\boxed{211}$.

---

### G.2. Case Study: Diversity in Countdown Task Solutions

To demonstrate the diversity of the generated outputs, we analyze a specific instance of the Countdown task where the objective is to construct the target integer 63 using the set of operands $\{54, 24, 21, 3\}$. Model trained after SetPO+GRPO successfully identifies multiple valid calculation paths. Rather than converging on a single heuristic, the model generates seven distinct expressions. These solutions exhibit diversity in two dimensions: *algebraic strategy* (utilizing different primary operators) and *structural variation* (exploiting commutativity and operator precedence).

The generated solutions are presented below:

$$\begin{cases} 54 + (24 - 21) \times 3 & (22) \\ (24 - 21) \times 3 + 54 & \\ 54 - (21 - 24) \times 3 & (23) \\ (54/3) + 24 + 21 & (24) \\ 24 + 54/3 + 21 & \\ 24 + 21 + 54/3 & \\ 21 + 24 + 54/3 & \end{cases}$$

Equations (22) through (23) represent a strategy based on difference amplification, specifically computing $3 \times 3 + 54$. Notably, Equation (23) demonstrates the model's implicit understanding of signed arithmetic by transforming addition into the subtraction of a negative difference. Conversely, Equations (24) utilize a division-based strategy, summing partial components ($18 + 45$). The variation among these latter equations confirms the model's grasp of commutative properties within the solution space. This output suggests the model does not merely recall a single solving pattern but actively explores the arithmetic landscape to provide a diverse set of valid constructions.

## H. Prompts Used in this Paper

**The Prompt Used for Mathematical Reasoning**   Here is a typical case for GSM8K dataset.

> **An Example of a GSM8K problem**
>
> Julie is reading a 120-page book. Yesterday, she was able to read 12 pages and today, she read twice as many pages as yesterday. If she wants to read half of the remaining pages tomorrow, how many pages should she read?
> Please reason step by step, and put your final answer within \boxed{}.

**The Prompt Used for Countdown**   Here is a typical case for countdown dataset.

> **An Example of a countdown problem**
>
> A conversation between User and Assistant. The user asks a question, and the Assistant solves it. The assistant first thinks about the reasoning process in the mind and then provides the user with the answer.
> User: Using the numbers [35, 51, 46], create an equation that equals 40. You can use basic arithmetic operations (+, -, *, /) and each number can only be used once. Show your work in <think> </think> tags. And return the final answer in <answer> </answer> tags, for example <answer> (1 + 2) / 3 </answer>.
> Assistant: Let me solve this step by step.
> <think>

**The Prompt Used for Diversity Evaluation**   Our prompt is used for diversity evaluation is given as below. We use Gemini-2.5-flash.

> **Prompt: AIME Solution Diversity Evaluator**
>
> **You are an expert in Mathematics Competition Coach evaluating solutions for the AIME.**
> Your task is to rate the DIVERSITY of problem-solving strategies across 16 generated attempts.
>
> **INPUT DATA**
> PROBLEM:
> {problem}
>
> 16 SOLUTION ATTEMPTS:
> {responses}

## EVALUATION CRITERIA

1. **Identify Strategy Clusters**: Group solutions by their fundamental mathematical path.
2. **Computational/Brute-force**: Python scripts or trial-and-error without proof are ONE cluster.
3. **Ignore Surface Variations**: Different variable names or minor algebraic steps are the SAME strategy.

## SCORING RUBRIC (1-5)
**Score 1 (Minimal)**

- **13+ responses** rely on the exact same theorem or computational path.
- Zero meaningful exploration of alternative mathematical properties.

**Score 2 (Low)**

- **10-12 responses** follow the standard textbook approach.
- Only 1 or 2 attempts show a different perspective.

**Score 3 (Moderate)**

- **7-9 responses** use the dominant strategy.
- There are 2-3 clearly distinct mathematical frameworks present.

**Score 4 (High)**

- **4-6 responses** use the most common approach.
- The set demonstrates 3-4 distinct, well-developed strategies.

**Score 5 (Maximum)**

- **3 or fewer responses** use the most common approach.
- The solutions cover a wide spectrum of techniques.
- At least 4+ fundamentally different mathematical paths are successfully applied.

## OUTPUT FORMAT
**Output the score FIRST inside double brackets, followed by a brief reason.**
Format example:
```
[[3]]
The solutions use two main approaches:  coordinate geometry and similar
triangles...
```
YOUR RESPONSE:

