# OpenReview forum: "SetPO: Set-Level Policy Optimization for Diversity-Preserving LLM Reasoning"
_ICML.cc/2026/Conference — ICML 2026 regular_

### Official Review · Reviewer_p1uw · 2026-03-02

**Soundness:** 2
**Presentation:** 3
**Significance:** 3
**Originality:** 3
**Overall Recommendation:** 4
**Confidence:** 3

**Summary:**

This paper proposes SetPO, an RL post-training algorithm that aims to increase the diversity of responses, therefore enabling the model to explore the solution space. SetPO adds a diversity score for each response to its advantage to incentivize different responses. The diversity score is computed based on the semantic similarity between the target response and the other responses in the same set. Intuitively, responses that are semantically farther from the others receive a higher diversity score.

The paper validates the method through both theoretical analysis of the diversity score and empirical experiments on math problems. The results show performance gains in both pass@1 and pass@k.

**Compliance With Llm Reviewing Policy:**

Affirmed.

**Key Questions For Authors:**

Could authors solve or explain the weaknesses mentioned above?

**Limitations:**

yes

**Strengths And Weaknesses:**

**Strengths**

1. Significance: the paper aims at solving a core problem in RL post-training, extending the models’ capability beyond the boundary of base models. The paper showcase noticeable gain in experiments.
2. Soundness: The claims are well supported by experiments and theoretical derivation of the method. The experiments are comprehensively conducted on mainstream math benchmarks and models of a wide range of sizes.
3. Presentation: the paper is well motivated and clear to follow.
4. Originality: This work provides new insights into solving the limits of RLVR, proposing a new way to boost the diversity of models’ responses. This differs from the prior works that leverage entropy bonus or target pass@K.

**Weaknesses**
1. (minor weakenss) The theoretical analysis does not provide any performance guarantee of the algorithm. It’s more like a justification or derivation of the algorithm design.
2. Presentation: Current description of the motivation experiment in Figure 1 is too concise. A separate paragraph of full introduction may be preferred.
3. Experiment: SetPO aims at solving the problem described in limit-of-RLVR [1]. To understand if SetPO solves the limits of RLVR, the following results would be appreciated to see: the results of pass@K with a larger K (e.g., 1024); comparison between SetPO and the base models (e.g., Qwen-series models in the paper) on pass@K, and observation of whether SetPO can achieve higher pass@K with a large K; perplexity analysis (Figure 6 in [1]).
4. In experiments, the performance of SetPO is compared with various standard algorithms (GRPO, DAPO, and GSPO). However, none of these methods aim at improving the output diversity. To truly demonstrate the benefits of SetPO, it is suggested to include prior related works (e.g., entropy-based regularization, trajectory-level semantic diversity measures) as baselines.

[1] Does Reinforcement Learning Really Incentivize Reasoning Capacity in LLMs Beyond the Base Model?

---

> ### Author Rebuttal · Authors · 2026-03-31
>
> Thank you for taking the time to review our manuscript. We greatly appreciate your valuable feedback. Below, we provide our responses to your comments:
>
> - Scope and role of the theoretical analysis (Weakness 1)
>
> The theory is intended to justify the diversity signal itself rather than provide a full end to end performance guarantee. SetPO defines a set level diversity functional, derives the exact finite set leave one out marginal decomposition, and proves anti redundancy and monotonicity properties. We believe this gives the algorithm a principled foundation, rather than introducing an ad hoc diversity regularizer.
>
> - The presentation of the toy example (Weakness 2)
>
> We will expand the description of Figure 1 in the revision. The toy model is a multi modal bandit with 12 discrete modes, of which only the first 4 are reward bearing and the remaining 8 are distractors. Each mode is assigned a fixed 50-dimensional embedding, and semantic similarity is measured by cosine similarity between these embeddings. There are multiple equally correct rewarding modes, under which vanilla GRPO tends to collapse onto a single correct mode, while SetPO preserves broader coverage.
>
> - Large $K$ evaluation and perplexity analysis (Weakness 3)
>
> We have included a direct comparison between the trained model and the original base model. Specifically, **Appendix F.2 (Figure 8)** compares the instruction tuned base model and SetPO+GRPO on AIME24 from Pass@1 through Pass@256, and SetPO+GRPO remains stronger across the full range. This supports our claim that the gains do not disappear as the sampling budget grows.
>
> We additionally follow the perplexity style diagnostic in [1]. Using the base model as the scorer, we compute the perplexity of responses generated by the base model, GRPO, and SetPO+GRPO on AIME24:
>
> |Generator|Mean PPL under base scorer|Std. PPL under base scorer|
> |---|---|---|
> |Base|1.386|0.335|
> |GRPO|1.343|0.261|
> |SetPO+GRPO|1.371|0.303|
>
> Under this diagnostic, GRPO exhibits both a lower mean perplexity and a substantially smaller standard deviation, which is consistent with a more concentrated response distribution. By contrast, SetPO+GRPO has a spread closer to the base model and substantially broader than GRPO. This is consistent with SetPO mitigating the concentration effect of vanilla GRPO and better preserving diversity.
>
> - Comparison with diversity-driven baselines (Weakness 4)
>
> To compare with diversity-driven baselines, we compare against **R1-Zero-Div** [2] in our original paper in **Table 1 and Table 2**. The evaluation is not restricted to standard RL baselines that do not focus on diversity. R1-Zero-Div is an entropy based method that encourages diversity through token level entropy style regularization, whereas SetPO defines diversity at the set level and assigns each trajectory an exact leave one out marginal contribution. We also add an additional comparison with Risk-RL from [3]. For a supplementary comparison, we train the same Qwen2.5-Math-7B backbone by closely following the setting in [3]. Since the Risk-RL work does not publicly release a complete end to end training codebase to our knowledge, we report the Risk-RL numbers from the original paper and compare them with our SetPO+GRPO results under the matched setting. The corresponding results are summarized below:
>
> |Method|AIME24 (Pass@1 / Pass@32)|AIME25 (Pass@1 / Pass@32)|MATH500 (Pass@1 / Pass@32)|
> |---|---|---|---|
> |Risk-RL (dapo17k)|34.2 / 65.8|18.7 / 40.7|80.4 / 94.8|
> |Risk-RL (math12k)|33.1 / 59.4|16.7 / 37.6|81.5 / 94.8|
> |SetPO+GRPO (dapo17k)|**35.3 / 67.9**|**19.3 / 44.9**|**82.1** / 94.8|
> |SetPO+GRPO (math12k)|**34.8 / 60.1**|**18.5 / 37.8**|**83.5** / **95.2**|
>
> These additional comparisons further validate the effectiveness of SetPO.
>
> [1] Yue, X., et al. Does Reinforcement Learning Really Incentivize Reasoning Capacity in LLMs Beyond the Base Model?
>
> [2] Yao, J., et al. Diversity-aware policy optimization for large language model reasoning. In Advances in Neural Information Processing Systems 38, 2025.
>
> [3] Jiang, Y., et al. Risk-Sensitive RL for Alleviating Exploration Dilemmas in Large Language Models
>
> We hope these updates and responses adequately address your concerns. If you have further questions or need additional clarifications, we would be happy to provide them.

---

> > ### Author Rebuttal · Reviewer_p1uw · 2026-04-01
> >
> > Thank authors for the reply. The authors have addressed most of my concerns. I suggest authors to implement baselines like R1-Zero-Div and Risk-RL and report full evaluation results (all 6 benchmarks) in the next version.
> >
> > Since I've already suggested acceptance, I'd keep my score unchanged.

---

> > > ### Author Response · Authors · 2026-04-03
> > >
> > > We sincerely thank the reviewer for the constructive suggestions and positive feedback, and we will incorporate additional baselines and report full results in the next version.

---

### Official Review · Reviewer_RGAE · 2026-03-04

**Soundness:** 3
**Presentation:** 3
**Significance:** 2
**Originality:** 3
**Overall Recommendation:** 4
**Confidence:** 4

**Summary:**

This paper proposes a variant of GRPO algorithm that adds trajectory-level semantic diversity to encourage diverse generations during RLVR process. More specifically, this work proposes diversity contribution which measures the marginal difference between the whole set and leave-one-out sample. On mathematical reasoning benchmarks, the proposed method achieves higher pass@1 and pass@k performance.

**Compliance With Llm Reviewing Policy:**

Affirmed.

**Final Justification:**

I thank the authors for their detailed responses and additional experiments. However, my concerns are not fully addressed:
1, to concerns with generalizability:
While I agree the task reward is computed using verifiable function, the similarity based rewards are also added to the advantage computation, which will affect the training process. As a result, my concern with the generalizability is not fully addressed.

2, to concerns with marginal improvements:
I thank the authors for pointing out that numbers in parentheses denote the bonus. My concern with marginal improvements upon baselines is addressed.

As a result, I decided to raise my score.

**Key Questions For Authors:**

1, In Table 1 and 2, Why were only the variances of SetPO method reported?
2, What were the embedding model(s) used in these experiments?

**Limitations:**

The performance of the proposed method would highly depend on the generality and performance of the embedding model.

**Strengths And Weaknesses:**

# Strength
1, This work measures the diversity via kernel method, which serves to capture trajectory-level and semantic-level diversities. This idea is novel and a good contribution to the community.

2, The the marginal difference is supported by theoretical analysis on its property.

3, The empirical results are good, showing that the proposed method could outperform baselines in pass@1 and pass@k.

# Weakness
1, It is not clear to me how the similarity is measured, e.g., what embedding model(s) has been used? Furthermore, the introduction of such embedding model diminishes the rigorousness of RLVR and raises concerns about their generalizability.

2, The proposed methods only marginally outperform baselines; also, the selected baselines are weak in that they do not consider improving pass@k at all. Adding more updated baselines, like [1] or other methods would significantly strengthen the work. Without such comparison to previous entropy based method, it is unclear what unique value the proposed diversity reward could bring.

[1] Risk-Sensitive RL for Alleviating Exploration Dilemmas in Large Language Models

---

> ### Author Rebuttal · Authors · 2026-03-31
>
> Thank you for taking the time to review our manuscript. We greatly appreciate your valuable feedback. Below, we provide our responses to your comments:
>
> - Embedding model choice and generalizability (Weakness 1 and Question 2)
>
> We measure similarity with a fixed pretrained embedding model, Qwen3-Embedding-0.6B. This does not diminish the rigor of RLVR, since the task reward itself remains fully rule-based and verifiable, while the component related to the embedding model is only an auxiliary diversity term added through $\hat A_i = \bar A_i + \lambda s_i$. Its influence is explicitly controlled by $\lambda$.
>
> Regarding the concern about generalizability, our models are trained only on the standard GSM8K training split, while evaluation is conducted on a broad set of mathematical reasoning benchmarks including MATH500, College Math, AMC23, AIME24, and AIME25. SetPO consistently improves over the baselines on these out of domain benchmarks, indicating that introducing the diversity term does not harm generalization.
>
> - On Table 1 and Table 2
>
> The number given in the parentheses denotes the bonus achieved by SetPO compared to the base method.
>
> - Comparison with diversity-driven baselines (Weakness 2)
>
> In our original submission, SetPO shows substantial improvements on challenging benchmarks such as AIME24 and AIME25, and the evaluation is not limited to baselines that do not focus on diversity. Specifically, Tables 1 and 2 compare SetPO against **R1-Zero-Div from [1]**, which incorporates an entropy based penalty to improve diversity in reasoning.
>
> SetPO does not rely on token level entropy style regularization. Instead, we define diversity at the set level and assign each trajectory a leave one out marginal contribution, directly rewarding rare reasoning paths. In addition, for a supplementary comparison with Risk-RL [2], we train the same Qwen2.5-Math-7B backbone by closely following the setting in [2]. Since the Risk-RL work does not publicly release a complete end to end training codebase to our knowledge, we report the Risk-RL numbers from the original paper and compare them with our SetPO+GRPO results under the matched setting. The corresponding results are summarized below:
>
> |Method|AIME24 (Pass@1 / Pass@32)|AIME25 (Pass@1 / Pass@32)|MATH500 (Pass@1 / Pass@32)|
> |---|---|---|---|
> |Risk-RL (dapo17k)|34.2 / 65.8|18.7 / 40.7|80.4 / 94.8|
> |Risk-RL (math12k)|33.1 / 59.4|16.7 / 37.6|81.5 / 94.8|
> |SetPO+GRPO (dapo17k)|**35.3 / 67.9**|**19.3 / 44.9**|**82.1** / 94.8|
> |SetPO+GRPO (math12k)|**34.8 / 60.1**|**18.5 / 37.8**|**83.5** / **95.2**|
>
> These additional comparisons further validate the effectiveness of SetPO. Across different comparison settings, SetPO consistently delivers stronger performance, showing that the proposed method is effective.
>
> [1] Yao, J., et al. Diversity-aware policy optimization for large language model reasoning. In Advances in Neural Information Processing Systems 38, 2025.
>
> [2] Jiang, Y., et al. Risk-Sensitive RL for Alleviating Exploration Dilemmas in Large Language Models
>
> We hope these updates and responses adequately address your concerns. If you have further questions or need additional clarifications, we would be happy to provide them.

---

> > ### Author Rebuttal · Reviewer_RGAE · 2026-04-03
> >
> > I thank the authors for their detailed responses and additional experiments. However, my concerns are not fully addressed:
> > 1, to concerns with generalizability:
> > While I agree the task reward is computed using verifiable function, the similarity based rewards are also added to the advantage computation, which will affect the training process. As a result, my concern with the generalizability is not fully addressed.
> >
> > 2, to concerns with marginal improvements:
> > I thank the authors for pointing out that numbers in parentheses denote the bonus. My concern with marginal improvements upon baselines is addressed.
> >
> > As a result, I decided to raise my score.

---

### Official Review · Reviewer_Szac · 2026-03-12

**Soundness:** 3
**Presentation:** 3
**Significance:** 2
**Originality:** 2
**Overall Recommendation:** 4
**Confidence:** 3

**Summary:**

The paper presents SetPO, an approach for set-level policy optimization aimed at enhancing diversity in reasoning capabilities of LLMs through reinforcement learning. The main contributions include the establishment of a mathematical framework for measuring diversity in outputs, leveraging group-based advantages to encourage diverse solutions, and demonstrating superior performance on various reasoning benchmarks compared to the relevant baselines.

**Compliance With Llm Reviewing Policy:**

Affirmed.

**Final Justification:**

The authors' rebuttal has sufficiently addressed all my concerns. Furthermore, in light of the authors' detailed response, as well as their replies to other reviewers—particularly the compelling experimental validations provided—I have decided to raise my Overall Recommendation score (3 -> 4).

**Key Questions For Authors:**

See Weakness.

**Limitations:**

The author does not provide or commit to making reproducible code publicly available.

**Strengths And Weaknesses:**

Strengths：

1. The paper has a clear motivation and rich theoretical analysis.
2. The paper is well-organized, and the topic it addresses is a current research hotspot. I think the proposed method holds potential practical application value.
3. Empirical evaluations show that SetPO, when combined with current mainstream baselines, consistently brings benefits.

Weaknesses：

1. Although the inclusion of the leave-one-out marginal contribution mechanism provides a deeper insight into diversity, it may introduce additional computational costs due to the reliance on an external embedding model for semantic similarity calculations. While the overhead is reported to be manageable, it could impact training throughput, especially in resource-constrained environments.
2. The parameter λ controlling the balance between task performance and diversity introduces complexity. It may require extensive tuning to achieve optimal performance, and the wrong choice of λ could either lead to subpar reasoning performance or a lack of necessary diversity, effectively limiting the algorithm's practical usability.
3. There is a lack of hyperparameter sensitivity analysis. I believe SetPO may be sensitive to hyperparameters; for example, λ is a very important hyperparameter, but I could not find any information about how λ is chosen and set. Additionally, I think there should be a sensitivity analysis regarding the important hyperparameters.

---

> ### Author Rebuttal · Authors · 2026-03-31
>
> Thank you for taking the time to review our manuscript. We greatly appreciate your valuable feedback. Below, we provide our responses to your comments:
>
> - Hyperparameter sensitivity and robustness  (Questions 2-3)
>
> We add a compact $\lambda$ ablation to address the concern regarding the sensitivity of $\lambda$. This study uses exactly the same setting as our main Qwen2.5-Math-7B experiments and varies only $\lambda$. We also include $\lambda=0$ as the baseline case without SetPO.
>
> ||0|0.05|0.1|0.3|0.5|0.7|1.0|
> |---|---|---|---|---|---|---|---|
> |Overall (SetPO+GRPO)|47.2|52.8|52.8|52.2|51.7|51.6|51.4|
> |Overall (SetPO+GSPO)|48.4|50.9|51.1|51.0|50.8|50.5|49.7|
> |Overall (SetPO+DAPO)|51.7|53.6|53.5|53.8|53.7|53.4|52.9|
>
> This experiment validates the robustness of our method with respect to $\lambda$. Within a reasonable range of $\lambda$, SetPO consistently surpasses the corresponding baselines. More broadly, $\lambda$ is the only additional hyperparameter introduced by SetPO itself. The remaining training hyperparameters follow the original GRPO/GSPO/DAPO settings and are kept unchanged in our experiments.
>
> - Computational cost (Question 1)
>
> We have discussed the computation cost in the original paper. **Appendix E** specifies the embedding model used by SetPO, and **Appendix Figure 9** reports the wall clock overhead of adding SetPO to the original algorithms. In our implementation, we use a pretrained embedding model, Qwen3-Embedding-0.6B, which is relatively small. Moreover, our current implementation is only a basic version and does not include any dedicated program level optimization for the embedding or reward computation pipeline. Even under this unoptimized setting, the additional wall clock time remains **below 10%**. Relative to the gains brought by SetPO, this overhead is acceptable.
>
> We hope these updates and responses adequately address your concerns. If you have further questions or need additional clarifications, we would be happy to provide them.

---

> > ### Author Rebuttal · Reviewer_Szac · 2026-04-04
> >
> > Thanks to the authors' rebuttal, my concerns have been adequately addressed. Furthermore, in light of the authors' detailed response, as well as their replies to other reviewers—particularly the compelling experimental validations provided—I have decided to raise my Overall Recommendation score.

---

### Official Review · Reviewer_TuBp · 2026-03-13

**Soundness:** 3
**Presentation:** 3
**Significance:** 3
**Originality:** 3
**Overall Recommendation:** 4
**Confidence:** 3

**Summary:**

This paper proposes SetPO (Set-Level Policy Optimization) to address the decline in output diversity during reinforcement learning for LLM reasoning. Departing from traditional token-level entropy, SetPO introduces a set-level diversity objective using kernelized similarity to measure trajectory-level richness. By employing a leave-one-out approach, the method estimates the marginal contribution of each trajectory to the overall set diversity, resulting in a plug-and-play reward shaping term compatible with algorithms like GRPO and DAPO. Theoretical analysis confirms the estimator’s monotonicity and anti-redundancy, while experiments across 1.5B to 32B models demonstrate that SetPO significantly improves reasoning accuracy while effectively preserving semantic diversity.

**Compliance With Llm Reviewing Policy:**

Affirmed.

**Key Questions For Authors:**

1. As stated in W.1, could the authors provide the ablation results on $\lambda$? And is this parameter sensitive to different experimental data? Besides, given that $s_i$ in Equation 4 can be negative, does this imply that the model might penalize correct but frequent reasoning strategies?
2. There are numerous other works that address the mode collapse issue in GRPO from the perspective of diversity, mostly the variations of DAPO. Could the authors kindly provide more results especially the diversity statistics in Table 3 on other diversity-driven GRPO methods like GAPO[1], ARPO[2]? How is Set-PO different from these methods?

[1] Anschel, Oron, et al. "Group-aware reinforcement learning for output diversity in large language models." Proceedings of the 2025 Conference on Empirical Methods in Natural Language Processing. 2025.
[2] Guanting Dong, et al. "Agentic Entropy-Balanced Policy Optimization." Proceedings of the ACM Web Conference 2026, 2026.

**Limitations:**

Yes

**Strengths And Weaknesses:**

Strengths:
1. The authors provide detailed derivations for the variational properties of the diversity function and the exact decomposition of LOO contributions.
2. The method is easy to implement and compatible with current mainstream algorithms.
3. The experiments cover models of multiple scales and are validated on mainstream math benchmarks like GSM8K and MATH500.

Weaknesses:
1. The paper may lack ablation studies on the $\lambda$ coefficient in the augmented advantage. The sensitivity of this coefficient should be discussed.
2. Addressing the diversity issue in GRPO is not a novel core insight[1]. This work primarily proposes a new way of estimating diversity but fails to sufficiently demonstrate its advantages over existing diversity-driven methods like DAPO. This lack of comparative depth weakens the overall novelty of the contribution.

[1] Yao, J., et al. Diversity-aware policy optimization for large language model reasoning. In Advances in Neural Information Processing Systems 38, 2025.

---

> ### Author Rebuttal · Authors · 2026-03-31
>
> Thank you for your detailed suggestions. Below, we provide our responses to your comments:
>
> - Hyperparameter sensitivity (W1 and Q1)
>
> We add a compact $\lambda$ ablation to demonstrate the effect of the hyperparameter $\lambda$. This ablation study is conducted under exactly the same setting as our main Qwen2.5-Math-7B experiments in the paper. The only change is the diversity coefficient $\lambda$. We also report $\lambda=0$ to recover the corresponding baseline without SetPO.
>
> ||0|0.05|0.1|0.3|0.5|0.7|1.0|
> |-|-|-|-|-|-|-|-|
> |Overall (SetPO+GRPO)|47.2|52.8|52.8|52.2|51.7|51.6|51.4|
> |Overall (SetPO+GSPO)|48.4|50.9|51.1|51.0|50.8|50.5|49.7|
> |Overall (SetPO+DAPO)|51.7|53.6|53.5|53.8|53.7|53.4|52.9|
>
> Moreover, following the setting of [1], we implement our algorithm on two additional datasets. Using the same $\lambda$ on both datasets, our algorithm outperforms Risk-RL proposed in [1].
>
> |Method|AIME24 (Pass@1 / Pass@32)|AIME25 (Pass@1 / Pass@32)|MATH500 (Pass@1 / Pass@32)|
> |-|-|-|-|
> |Risk-RL (dapo17k)|34.2 / 65.8|18.7 / 40.7|80.4 / 94.8|
> |Risk-RL (math12k)|33.1 / 59.4|16.7 / 37.6|81.5 / 94.8|
> |SetPO+GRPO (dapo17k)|**35.3 / 67.9**|**19.3 / 44.9**|**82.1** / 94.8|
> |SetPO+GRPO (math12k)|**34.8 / 60.1**|**18.5 / 37.8**|**83.5** / **95.2**|
>
> These experiments validate the robustness of our method with respect to $\lambda$.
>
> - Interpretation of negative leave one out values (Q1)
>
> $s_i < 0$ means that a trajectory is relatively redundant within the sampled set. Therefore, SetPO indeed downweights a correct but frequent reasoning pattern relative to another correct trajectory that contributes new set level information. However, this does not change whether the trajectory is correct: in $\hat{A}_i = \bar{A}_i + \lambda s_i$, correctness is still determined by the original reward, while $s_i$ only measures the set level contribution on diversity.
>
> - Comparison with existing methods (W2)
>
> We would like to first clarify that the diversity-driven method in [2] is named **R1-Zero-Div**, which we have compared against in the original paper. In **Table 1** and **Table 2**, we compare SetPO with R1-Zero-Div. R1-Zero-Div encourages diversity through token level entropy based regularization. SetPO directly rewards rare reasoning paths to avoid mode collapse. Quantitatively, SetPO also consistently outperforms R1-Zero-Div on **all overlapping benchmarks**. For Qwen2.5-Math-7B, SetPO+GRPO improves over R1-Zero-Div by +1.6 on GSM8K and +3.9 on MATH500. For Qwen2.5-Math-1.5B, SetPO+GRPO improves by +3.7 on GSM8K, +2.5 on MATH500, and +1.8 on College Math.
>
> - Relation to GAPO [3] and ARPO [4] (Q2)
>
> Methodologically, SetPO is different from GAPO and AEPO (the main algorithm proposed in [4] is named AEPO in the original paper). SetPO is a set level shaping method defined directly over sampled trajectories, where each trajectory receives a leave one out marginal contribution. Hence, it aims to preserve the semantic diversity of the sampled trajectory set. GAPO focuses on balancing frequencies over multiple valid final completions, which is limited to settings where all different solutions are completely known. AEPO, by contrast, analyzes exploration at the rollout level in agentic trajectories. These methods target different aspects of diversity and have a different scope from SetPO.
>
> For the diversity driven methods, beyond the Countdown results in Table 3, we also include a separate diversity analysis on AIME24. For each problem, we sample 16 solutions and use Gemini-2.5-Flash to assess the number of distinct underlying mathematical solution paths while ignoring surface form variation. We believe this is a suitable protocol for general mathematical reasoning, where the final answer is usually unique and exact mode counting is unavailable. AEPO likewise evaluates diversity at the trajectory level in its own agentic setting. By contrast, Table 3 measures a more specific notion of diversity that is unique to Countdown: because each sampled expression can be exactly verified, one can explicitly count how many distinct correct solutions are found for the same problem. GAPO, however, mainly encourages balanced sampling over multiple valid final completions through a frequency aware group reward, which is less suitable for mathematical tasks where there is typically a unique correct final answer. For this reason, directly extending the Table 3 statistics to these methods would not yield a directly comparable evaluation.
>
> [1] Jiang, Y., et al. Risk-Sensitive RL for Alleviating Exploration Dilemmas in Large Language Models
> [2] Yao, J., et al. Diversity-aware policy optimization for large language model reasoning.
> [3] Anschel, Oron, et al. Group-aware reinforcement learning for output diversity in large language models.
> [4] Dong, G., et al. Agentic Entropy-Balanced Policy Optimization.
>
> We hope these responses adequately address your concerns. If you have further questions or need additional clarifications, we would be happy to provide them.

---

> > ### Author Rebuttal · Reviewer_TuBp · 2026-04-04
> >
> > Thanks for the additional results and clarifications. However, my main concerns are still not fully resolved.

---

### Decision · Program_Chairs · 2026-04-30

**Decision:**

Accept (regular)

**Comment:**

The paper aims to address the challenge of output diversity drop during the RL procedure of LLM reasoning. All reviewers agreed that this paper is technically sound and well motivated. Several concerns are raised during rebuttal, including the comparison with previous diversity-driven methods (Reviewer TuBp and Reviewer RGAE) and the sensitivity of the proposed method (Reviewer TuBp and Reviewer Szac). In rebuttal, the authors successfully addressed these concerns with added ablations and comparisons to related baselines. Therefore, I recommend accptance.